# 3d gravity from Virasoro TQFT: Holography, wormholes and knots

**Scott Collier[1,2]⋆, Lorenz Eberhardt[3,4]† and Mengyang Zhang[5]‡**

**1** Princeton Center for Theoretical Science,
Princeton University, Princeton, NJ 08544, USA
**2** Center for Theoretical Physics, Massachusetts Insitute of Technology,
Cambridge, MA 02139, USA
**3** Institute for Advanced Study, Einstein Drive, Princeton, NJ 08540, USA
**4** Institute for Theoretical Physics, University of Amsterdam,
PO Box 94485, 1090 GL Amsterdam, The Netherlands
**5** Joseph Henry Laboratories, Princeton University, Princeton, NJ 08544, USA

⋆ sac@mit.edu , † l.eberhardt@uva.nl , ‡ mengyang@princeton.edu

## Abstract

We further develop the description of three-dimensional quantum gravity with negative cosmological constant in terms of Virasoro TQFT formulated in our previous paper [1]. We compare the partition functions computed in the Virasoro TQFT formalism to the semiclassical evaluation of Euclidean gravity partition functions. This matching is highly non-trivial, but can be checked directly in some examples. We then showcase the formalism in action, by computing the gravity partition functions of many relevant topologies. For holographic applications, we focus on the partition functions of Euclidean multi-boundary wormholes with three-punctured spheres as boundaries. This precisely quantifies the higher moments of the structure constants in the proposed ensemble boundary dual and subjects the proposal to thorough checks. Finally, we investigate in detail the example of the figure eight knot complement as a hyperbolic 3-manifold. We show that the Virasoro TQFT partition function is identical to the partition function computed in Teichmüller theory, thus giving strong evidence for the equivalence of these TQFTs. We also show how to produce a large class of manifolds via Dehn surgery on the figure eight knot.

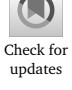

# 1 Introduction

Three-dimensional quantum gravity with negative cosmological constant has proven to be one of the most interesting and productive toy models of quantum gravity. The major outstanding problems are to fully solve the theory from first principles and firmly establish a holographic correspondence for the theory. There has been major progress on both fronts over the last few years; see [1–7] and [8–14], respectively.

In our previous paper [1], we developed a formalism that computes the gravity partition function algorithmically on a background of fixed (on-shell) topology. This fixes the contributions of hyperbolic three-manifolds to the gravitational path integral and represents a large step towards a complete solution of the theory directly from the bulk. The next step would involve performing the sum over all three-dimensional topologies that appear in the gravitational path integral, which may also require suitable non-perturbative or off-shell contributions. While our previous paper developed the formalism in terms of the Virasoro TQFT, this paper gives several interesting applications that exemplify its practical utility and should be viewed as a natural continuation of [1].

On the holographic side, a consistent picture is emerging that the gravitational path integral computes certain universal statistical features in a putative ensemble of holographic 2d CFTs. While the full non-perturbative definition of such an ensemble is still not settled, this perspective makes very concrete predictions that can be quantitatively matched between the bulk and boundary. In this paper we will study partition functions of Virasoro TQFT on multi-boundary wormholes to exemplify the extent to which the gravitational path integral precisely captures universal statistics of CFT data, transcending the Gaussian approximation of [10].

We will assume that the reader is acquainted with the concepts introduced in [1], but now recall some key features. As suggested by holography, the Hilbert space of 3d quantum gravity is spanned by the left- and right-moving Virasoro conformal blocks on the spatial surface $\Sigma$. This factorization of the Hilbert space allows one to consider, say, only the left-movers as a

fundamental building block.[1] A key ingredient in the proposal of [1] is an explicit form of the inner product on this conformal block Hilbert space. Important for the consistency of this structure is the fact that the conformal blocks transform among each other under crossing transformations and as such the Hilbert space carries a unitary action under crossing transformations. There are remarkably explicit expressions for the crossing transformation in terms of the Ponsot-Teschner fusion kernel $\mathbb{F}$ and the modular crossing kernel $\mathbb{S}$ [15–18]. Since the Hamiltonian in gravity vanishes, the theory can be viewed as a TQFT on a background topology. This data completely specifies the TQFT that we called Virasoro TQFT in [1]. The TQFT partition function on a fixed topology can be computed via surgery techniques similarly to Chern-Simons theory. The Virasoro TQFT partition function then immediately leads to the full 3d gravity partition function via the following formula, valid for all hyperbolic three-manifolds

$$Z_{\text{grav}}(M) = \sum_{\gamma \in \text{Map}(\partial M)/\text{Map}(M, \partial M)} |Z_{\text{Vir}}(M^\gamma)|^2 \ . \tag{1.1}$$

Here we sum over all images of the manifold $M$ under the boundary mapping class group $\text{Map}(\partial M)$, which is part of the sum over topologies, modulo the bulk mapping class group $\text{Map}(M, \partial M)$, which is gauged in gravity.

We start in Section 2 by analyzing the gravity partition functions as computed in Virasoro TQFT and their relation to the semiclassical evaluation of the gravitational path integral. Comparing the two expressions leads to the (refined) volume conjecture that we already mentioned in [1] and discuss further here. We also discuss the existence of non-isomorphic hyperbolic manifolds with identical Virasoro TQFT partition functions. This in particular implies that the gravitational path integral is not powerful enough to detect the topology of hyperbolic manifolds.

We then discuss examples that are relevant for the holographic description of 3d gravity in Section 3. We focus on a class of manifolds obtained by removing three-punctured spheres from $S^3$ and connecting the boundaries appropriately with Wilson lines. They compute holographically the higher moments of the structure constants in the proposed ensemble description of the boundary dual. We find that the partition functions may be computed using diagrammatic rules that are simply the $q$-deformations of the rules for the computation of disk partition functions in JT gravity + matter [19–21]. When projecting the Wilson lines on a disk, one associates a Virasoro 6j-symbol to every crossing of lines and one integral to every loop formed by the internal lines, see eqs. (3.63) and (3.64) for the precise formulae. We also use Virasoro TQFT to compute the gravity partition function on a class of contributions to the single-boundary gravitational path integral that are not handlebodies. These non-handlebody instantons are formed by quotients of the two-boundary Euclidean wormhole and we find that the gravity partition function is related to the partition function of Liouville CFT on a particular non-orientable surface.

Finally, we consider the example of the figure eight knot complement in Section 4, which constitutes one of the simplest examples of a hyperbolic manifold with no asymptotic boundary. We compute its Virasoro TQFT partition function from a variety of perspectives and demonstrate that it agrees with the partition function computed in an a priori different TQFT known as Teichmüller TQFT. This lends strong credence to the equivalence of the two theories, even though Virasoro TQFT provides a far more convenient framework for holographic applications. We also illustrate the procedure of Dehn surgery on the figure eight knot, which leads to the gravity partition function on a whole family of hyperbolic three-manifolds whose volume accumulates to that of the figure eight knot complement.

---

[1]The left- and right-movers are entangled only by the sum over topologies in the gravitational path integral.

## 2 Structural properties of Virasoro TQFT

We start by discussing the relation of the formulation of gravity in terms of Virasoro TQFT and the semiclassical gravity path integral. Comparing the two leads to the volume conjecture and we discuss various consequences for the volumes of hyperbolic 3-manifolds, conformal blocks and one-loop determinants. We then also explain some of the consistency conditions of the Virasoro TQFT. Such consistency conditions are all implied by the consistency of the mapping class representation on the initial value surface, but often the three-dimensional viewpoint is much more powerful.

### 2.1 Volume conjecture

We already stated the (refined) volume conjecture in [1], but it will play a much more prominent role in the present paper. By comparing the usual metric approach of 3d gravity and the Virasoro TQFT approach, one obtains the following prediction for the semiclassical expansion of partition functions:

$$
|Z_{\text{Vir}}(M)|^2 = e^{-\frac{c}{6\pi}\text{vol}(M)} \left[ \prod_{\gamma \in \mathcal{P}} \prod_{m=2}^{\infty} \frac{1}{|1-q_\gamma^m|^2} + \mathcal{O}(c^{-1}) \right] . \tag{2.1}
$$

Here we used that the gravity tree-level action is $\frac{c}{6\pi}\text{vol}(M)$, where $\text{vol}(M)$ is the volume of the hyperbolic manifold. We also used the explicit form of the one-loop determinant as computed in [3]. This explicit formula for the one-loop determinant is valid for hyperbolic manifolds without defects that can be written as $\mathbb{H}^3/\Gamma$ for a so-called Kleinian group $\Gamma$. In case $M$ has defects, the volume conjecture should still hold, but there is no known general formula for the one-loop determinant. We recall that $\mathcal{P}$ denotes the set of all primitive geodesics on the three-manifold in question. Alternatively, we can think of $\mathcal{P}$ as the set of primitive conjugacy classes in the Kleinian group $\Gamma$ (i.e. conjugacy classes that are not powers of other conjugacy classes) and also identify the conjugacy class of $\gamma$ with the conjugacy class of $\gamma^{-1}$, since this corresponds to orientation reversal of the corresponding geodesic. We could of course extend the matching to higher loop order, but will restrict here to the tree-level and one-loop piece.

We refer to this equation as the refined volume conjecture, since the classical volume conjecture is the corresponding statement for the tree-level term in the $\frac{1}{c}$-expansion [22]. The relation (2.1) should also hold in the presence of boundaries, in which case the volume of the hyperbolic manifold is the renormalized volume [23].

### 2.2 The volume of hyperbolic tetrahedra

Let us explain one of the simplest non-trivial instances of the volume conjecture in more detail. Consider a single hyperbolic tetrahedron as in Figure 1 with dihedral angles $\theta_i$ specifying the angle between the two faces meeting at the edge.

The dihedral angles have to satisfy rather complicated conditions for such a hyperbolic tetrahedron to exist. We can take two identical such hyperbolic tetrahedra and identify them along the corresponding faces. This leads to a topological three-sphere with conical defects running in the form of the tetrahedron through it. The hyperbolic tetrahedron is specified by the dihedral angles $\theta_j \in (0, \pi)$ spanned by the two faces meeting at an edge. Upon gluing two tetrahedra, the conical defect angle becomes $2\pi - 2\theta_j$. Semiclassically, the relation between

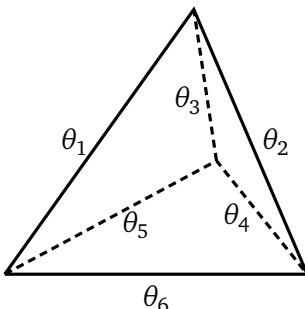

Figure 1: Tetrahedron with dihedral angles specified.

defect angles $\alpha_j$ and Liouville momentum reads[2] [24]

$$P_j = \frac{i\alpha_j}{4\pi b} \sim \frac{iQ}{2} - \frac{i\theta_j}{2\pi b}. \tag{2.3}$$

This is the hyperbolic three-manifold that we use for the volume conjecture. Some extra care is necessary to correctly normalize the vertices. We observe that the renormalized volume of the Euclidean wormhole of the form $\Sigma_{0,3} \times [0,1]$ exactly vanishes.[3] Since it evaluates to the Liouville structure constant $C_0(P_1, P_2, P_3)$ in the Virasoro TQFT, this means that for the purposes of the volume conjecture, we should define a juncture with a normalization constant $C_0(P_1, P_2, P_3)^{-\frac{1}{2}}$ as follows

$$\frac{1}{\sqrt{C_0(P_1, P_2, P_3)}} \times \quad P_2 \; \text{---} \; \bigcirc \; \overset{P_1}{\underset{P_3}{\diagdown}} \quad . \tag{2.4}$$

We compute the TQFT partition function $Z_{\text{Vir}}$ of this tetrahedral configuration in Section 3.2. The result is given by

$$Z_{\text{Vir}}\left( \vphantom{\int} \triangle \right) = \rho_0(P_6)^{-1} C_0(P_1, P_2, P_3) C_0(P_3, P_4, P_5) \mathbb{F}_{P_3, P_6}\begin{bmatrix} P_4 & P_2 \\ P_5 & P_1 \end{bmatrix} \tag{2.5}$$

$$= \sqrt{C_0(P_1, P_2, P_3) C_0(P_1, P_5, P_6) C_0(P_2, P_4, P_6) C_0(P_3, P_4, P_5)}$$
$$\times \begin{Bmatrix} P_1 & P_2 & P_3 \\ P_4 & P_5 & P_6 \end{Bmatrix}. \tag{2.6}$$

The symbol $\begin{Bmatrix} P_1 & P_2 & P_3 \\ P_4 & P_5 & P_6 \end{Bmatrix}$ is the crossing kernel for sphere four-point function conformal blocks in the Racah-Wigner normalization [17], which we also call the Virasoro 6j-symbol. It has the correct tetrahedral symmetry as required by the picture, where the vertices of the tetrahedron are formed by $(P_1, P_2, P_3)$, $(P_1, P_5, P_6)$, $(P_2, P_4, P_6)$ and $(P_3, P_4, P_5)$.

---

[2]Here we are adopting the standard notation from Liouville theory for the central charge and conformal weights:

$$c = 1 + 6(b + b^{-1})^2 = 1 + 6Q^2, \qquad \Delta_j = \frac{c-1}{24} + P_j^2. \tag{2.2}$$

[3]Here and throughout we use $\Sigma_{g,n}$ to refer to a Riemann surface of genus $g$ with $n$ punctures. So here $\Sigma_{0,3}$ corresponds to the three-punctured sphere.

Thus the prediction of the volume conjecture is now that

$$2\,\mathrm{vol}(\eta_1, \eta_2, \eta_3, \eta_4, \eta_5, \eta_6) = -2\pi \lim_{b \to 0} b^2 \log \left\{ \begin{matrix} \frac{iQ}{2} - \frac{i\eta_1}{2\pi b} & \frac{iQ}{2} - \frac{i\eta_2}{2\pi b} & \frac{iQ}{2} - \frac{i\eta_3}{2\pi b} \\ \frac{iQ}{2} - \frac{i\eta_4}{2\pi b} & \frac{iQ}{2} - \frac{i\eta_5}{2\pi b} & \frac{iQ}{2} - \frac{i\eta_6}{2\pi b} \end{matrix} \right\}, \qquad (2.7)$$

where the volume on the left hand side is the volume of the hyperbolic tetrahedron specified by the dihedral angles $\eta_j$.

One can evaluate the integral in the defining formula for the crossing kernel via saddle-point approximation in this limit and confirm that it agrees with the volume formula for a hyperbolic tetrahedron. This was done in [17], but the volume conjecture gives a conceptual derivation of that fact.

We also mention that the Virasoro crossing kernel has the following Regge symmetry [25, 26]

$$\mathbb{F}_{P_3, P_6} \begin{bmatrix} P_4 & P_2 \\ P_5 & P_1 \end{bmatrix} = \mathbb{F}_{P_3, P_6} \begin{bmatrix} \frac{1}{2}(P_2 + P_4 + P_5 - P_1) & \frac{1}{2}(P_1 + P_2 + P_4 - P_5) \\ \frac{1}{2}(P_1 + P_4 + P_5 - P_2) & \frac{1}{2}(P_1 + P_2 + P_5 - P_4) \end{bmatrix}. \qquad (2.8)$$

This implies via the volume conjecture that the volume of a hyperbolic tetrahedron is invariant under the replacement

$$\theta_1 \to \tfrac{1}{2}(\theta_1 + \theta_2 + \theta_4 - \theta_5), \qquad\qquad \theta_5 \to \tfrac{1}{2}(-\theta_1 + \theta_2 + \theta_4 + \theta_5), \qquad (2.9)$$

$$\theta_2 \to \tfrac{1}{2}(\theta_1 + \theta_2 + \theta_5 - \theta_4), \qquad\qquad \theta_4 \to \tfrac{1}{2}(\theta_1 - \theta_2 + \theta_4 + \theta_5), \qquad (2.10)$$

with $\theta_3$ and $\theta_6$ unchanged. This property is very non-trivial to see geometrically and giving a direct proof of it is rather hard.

## 2.3 Volume conjecture for handlebodies

**Semiclassical vacuum blocks.** Let us apply the volume conjecture in the form (2.1) to a handlebody. Recall that the Virasoro TQFT partition function on a genus-$g$ handlebody $S\Sigma_g$ evaluates to the vacuum Virasoro conformal block,

$$Z_{\mathrm{Vir}}(S\Sigma_g) = \vcenter{\hbox{}} \qquad (2.11)$$

where we drew a genus-2 surface for concreteness. As such the volume conjecture (2.1) gives the semiclassical expansion of vacuum blocks,

$$\vcenter{\hbox{}} \sim e^{-\frac{c}{12\pi}\mathrm{vol}(S\Sigma_g)} \prod_{\gamma \in \mathcal{P}(\Gamma_g)} \prod_{m=2}^{\infty} \frac{1}{1 - q_\gamma^m}. \qquad (2.12)$$

Here, $\mathrm{vol}(S\Sigma_g)$ is the in general complex volume of the handlebody.[4] Such a semiclassical expansion of the conformal blocks is familiar from 2d CFT, where the leading term is called the semiclassical conformal block [27–29], but to our knowledge there is no general CFT derivation of the one-loop determinant, and even direct derivations of the leading term are somewhat limited. The group $\Gamma_g \subset \mathrm{PSL}(2, \mathbb{C})$ that appears in the one-loop determinant is the Schottky group of the corresponding handlebody.

---

[4]In general to write this formula only for a chiral half (which goes beyond the volume conjecture (2.1)), we also need to assign an imaginary part to the volume which is known as the Chern-Simons invariant.

The Virasoro TQFT approach gives a simple derivation of this fact. It also shows that the semiclassical block is nothing else than the volume of the corresponding handlebody. It was shown in [30] that this volume is identified with the on-shell value of the Liouville action as defined by Takhtajan and Zograf [31],

$$S_{\mathrm{L}}(\Sigma_g) = -4 \operatorname{Re} \mathrm{vol}(\mathsf{S}\Sigma_g). \tag{2.13}$$

Defining the on-shell Liouville action requires one to pick a conformal block channel. The Virasoro TQFT also makes a prediction about the order one term in the semiclassical expansion.

**One-loop determinant.** Let us recall the formula derived in [32] for the holomorphic factorization of the Laplacian on a Riemann surface. We have

$$\frac{\det' \Delta_2}{\det N_2} = c_g \, e^{-\frac{13}{12\pi} S_{\mathrm{L}}(\Sigma_g)} \left| (1-q_1)^2 (1-q_2) \prod_{\gamma \in \mathcal{P}(\Gamma_g)} \prod_{m=2}^{\infty} (1-q_\gamma^m)^2 \right|^2, \tag{2.14}$$

for some constant $c_g$ independent of the moduli. It depends on the renormalization scheme used to define the determinant $\det' \Delta_2$. Here $\Delta_2$ is the Laplacian acting on holomorphic quadratic differentials on the surface $\Sigma$ and the prime indicates that we removed the zero modes. $\det N_2$ is the determinant of $\langle \varphi_j \,|\, \varphi_k \rangle$ and $\{\varphi_j\}_{j=1,\dots,3g-3}$ is a natural basis of holomorphic quadratic differentials as defined in [32]. We also denoted $q_j = q_{\gamma_j}$ with $\gamma_1, \dots, \gamma_g$ the $g$ free generators of the Schottky group. The perhaps unnatural seeming factor $(1-q_1)^2(1-q_2)$ appears because of the specific way in which $\varphi_j$ is defined and is a result of fixing the $\mathrm{PSL}(2,\mathbb{C})$ conjugacy freedom for the Schottky group. Thus we have

$$\left| \vphantom{\Bigg|} \quad\quad\quad\quad\quad\quad\quad\quad\quad\quad\quad \right|^2 \sim c_g' \, \frac{e^{-\frac{c-13}{6\pi} \mathrm{vol}(\mathsf{S}\Sigma_g)}}{\sqrt{\det' \Delta_2}} \times |1-q_1|^2 |1-q_2| \sqrt{\det N_2}. \tag{2.15}$$



This tells us that the one-loop partition function is exactly the inverse square root of the partition function of a $bc$-ghost system with a particular choice of ghost insertions.

**Explicit check.** Here we explicitly check the one-loop refinement of the volume conjecture for handlebodies in a simple example, perturbatively in the moduli of the Riemann surface in an expansion about a pinching limit. Consider for concreteness a genus-two Riemann surface formed by plumbing two two-holed disks $D_1$ and $D_2$:

$$D_1 = \{z_1 \in \mathbb{C} \,|\, r_1 < |z_1| < r_3, \, |z_1 - 1| > r_2\}, \tag{2.16a}$$

$$D_2 = \{z_2 \in \mathbb{C} \,|\, \tilde{r}_1 < |z_2| < \tilde{r}_3, \, |z_2 - 1| > \tilde{r}_2\}. \tag{2.16b}$$

Gluing the boundaries of the disks according to the following inversion map prepares a disk with three holes

$$|z_2| = \tilde{r}_3 : \quad z_2 \sim \frac{1}{p_1 z_1}, \quad \text{with} \quad |p_1| = \frac{1}{r_3 \tilde{r}_3}. \tag{2.17}$$

The remaining identifications are

$$|z_2| = \tilde{r}_1 : \quad z_2 \sim \frac{p_3}{z_1}, \quad\quad \text{with} \quad |p_3| = r_1 \tilde{r}_1, \tag{2.18a}$$

$$|z_2 - 1| = \tilde{r}_3 : \quad z_2 - 1 \sim \frac{p_2}{z_1 - 1}, \quad \text{with} \quad |p_2| = r_2 \tilde{r}_2. \tag{2.18b}$$

The complex plumbing parameters $p_i$ parameterize the moduli of the Riemann surface, with the $p_i \to 0$ limit a pinching locus in which the surface is realized by gluing two spheres along long narrow tubes. The corresponding Virasoro conformal blocks may then straightforwardly be computed as an expansion in powers of the plumbing parameters $p_i$, see for example [33] for details.

This parameterization of the genus-two Riemann surface is clearly equivalent to the Schottky parameterization, in which one realizes the Riemann surface $\Sigma_g$ as a quotient of the form

$$\Sigma_g = (\mathbb{C} \cup \{\infty\} - \Lambda)/\Gamma. \tag{2.19}$$

Here $\Gamma = \langle \gamma_1, \dots, \gamma_g \rangle$ is the Schottky group, which is a free group generated by the loxodromic elements $\gamma_1, \dots, \gamma_g$ of PSL$(2, \mathbb{C})$, and $\Lambda$ is the limit set of the action of $\Gamma$. The generators $\gamma_i$ act on the Riemann sphere by Möbius transformation. In our example of the genus-two Riemann surface formed by plumbing two-holed disks as above, the generators of the Schottky group may be taken to be

$$\gamma_1(z) = p_1 p_3 z, \qquad \gamma_2(z) = \frac{(1 - p_2)z - 1/p_1}{z - 1/p_1}. \tag{2.20}$$

Each generator $\gamma$ is conjugate to diag$(q_\gamma^{1/2}, q_\gamma^{-1/2})$, with $|q_\gamma| < 1$. Here we have

$$q_{\gamma_1} = p_1 p_3, \qquad q_{\gamma_2} = \frac{1 - p_1 + p_1 p_2 - \sqrt{1 - 2p_1(1 + p_2) + p_1^2(1 - p_2)^2}}{1 - p_1 + p_1 p_2 + \sqrt{1 - 2p_1(1 + p_2) + p_1^2(1 - p_2)^2}}. \tag{2.21}$$

We are now in a position to directly compare the perturbative expansion of the $c \to \infty$ limit of the genus-two Virasoro vacuum block as parameterized in the plumbing frame above[5] with that of the gravity one-loop determinant on the genus-two handlebody (2.12). We find

$$\left. \vcenter{\hbox{}} \right|_{c \to \infty} = \prod_{\gamma \in \mathcal{P}(\Gamma)} \prod_{m=2}^{\infty} \frac{1}{1 - q_\gamma^m} \tag{2.22}$$

$$= 1 + p_1^2 p_2^2 + p_2^2 p_3^2 + p_3^2 p_1^2 + 4(p_1^3 p_2^2 + p_2^3 p_3^2 + p_2^2 p_3^3) + \dots \tag{2.23}$$

On the left-hand side we evaluate the genus-two identity block in the plumbing frame perturbatively in the moduli by brute force, and on the right-hand side we evaluate the one-loop determinant by taking the product over primitive conjuacy classes of the Schottky group. We have verified the agreement between these two expressions up to total degree 12 in the expansion in the plumbing parameters.

## 2.4 Mutations of hyperbolic manifolds

One may ask whether $Z_{\text{Vir}}$ is a perfect invariant of a hyperbolic three-manifold, or, in other words, is $Z_{\text{Vir}}$ powerful enough to distinguish any two hyperbolic manifolds? As in Chern-Simons theory, the answer to this question is negative. There exist non-isometric hyperbolic three-manifolds $M_1$ and $M_2$ with $Z_{\text{Vir}}(M_1) = Z_{\text{Vir}}(M_2)$. The reason for this is a general operation known as mutation.

There are different kinds of mutations, these are all relatively subtle operations that go undetected by most knot invariants, including the Virasoro TQFT partition function $Z_{\text{Vir}}$.[6] Let us first explain the classical example of knot mutations. Consider a region of a knot in which

---

[5]As explained in [33], in the plumbing frame the $c \to \infty$ limit of the Virasoro blocks is actually finite; in other words, the corresponding Liouville action vanishes. The $c \to \infty$ limit of the vacuum block as computed in (2.12) plays an important role in determining the seed of the recursive representation of arbitrary Virasoro blocks.

[6]In Virasoro TQFT, we think of a knot as a defect inserted in the three-sphere $S^3$ that is knotted appropriately.

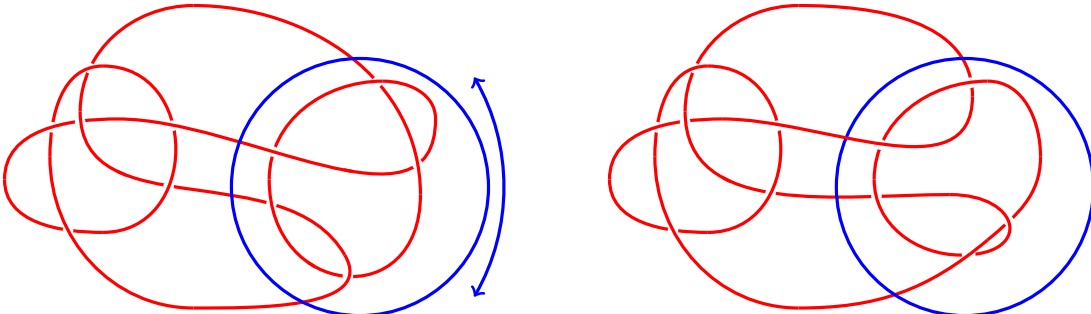

Figure 2: The Conway knot and its mutant, the Kinoshita-Teresaka knot. They are not equivalent, but the value of $Z_{\text{Vir}}$ is the same.

two strands enter and two strands exit. The path integral over this region gives a state in the Hilbert space of the four-punctured sphere where all the four labels are identical (since we considering a knot associated to a single Virasoro representation). Thus, whatever the resulting state is, it can be expanded in terms of $s$-channel conformal blocks. However, any $s$-channel conformal block with identical external labels is invariant under a $\mathbb{Z}_2 \times \mathbb{Z}_2$ symmetry group generated by rotations around the $x$ and $y$ axis as follows:

$$\tag{2.24}$$

the composition of which yields a rotation by 180 degrees. It thus follows that the Virasoro TQFT partition function on the excised four-punctured sphere is invariant under the same symmetry operations. In particular, this means that one can cut the four-punctured sphere with a tangle inside, apply one of these symmetry operations, and then reglue the tangle. This leads in general to an inequivalent knot, but the difference is not detectable by computing $Z_{\text{Vir}}$. A famous example of a mutant hyperbolic knot pair is the Conway knot and the Kinoshita-Terasaka knot shown in Figure 2. In particular, since for this example, the mapping class group of both knot complements is trivial, the gravitational path integral on the Conway knot and the Kinoshita-Terasaka knot is exactly the same and the gravitational path integral is hence not a sufficiently refined observable to be able to detect the topology of all hyperbolic three-manifolds.

Via the volume conjecture (2.1), this implies in particular that mutant knot complements have the same hyperbolic volume. This result is known in the math literature [34], but the present discussion makes it tautological. More surprisingly, the refined volume conjecture (2.1) also implies that the corresponding manifolds have the same one-loop determinants.

Using the same techniques of Virasoro TQFT, one can also show that the geodesics fully inside or outside the cutting surface have the same length.[7] Thus the length spectra of two mutant manifolds partially coincide. However, the length spectrum in general differs as one can see by an explicit computation using the software SnapPy [35]. We display in Table 1 the low-lying length spectrum on the Conway knot and the Kinoshita-Teresaka knot. Thus even though the geodesic length spectrum determines the one-loop determinant (we have $q_\gamma = e^{-\ell_\gamma}$) and the one-loop determinants agree, the length spectrum is in general different.

There are other versions of mutations. We can consider any embedded surface in $M$ with a special symmetry such as the four-punctured sphere above. Cutting $M$ along such a surface, applying the symmetry and regluing leads to a mutated manifold. For example, we can cut

---

[7]This is based on the observation that inserting Wilson lines with degenerate Virasoro representations measure the geodesic length in the classical limit [26].

Table 1: The low-lying length-spectrum of primitive geodesics on the complement of the Conway and the Kinoshita-Teresaka knot. The imaginary part encodes the holonomy around the geodesic. The length spectrum partially agrees. All geodesics have multiplicity 1 since the two manifolds have trivial isometry groups.

| Conway | Kinoshita-Teresaka |
|---|---|
| $1.044 + 2.327i$ | $1.044 + 2.327i$ |
| $1.152 - 2.266i$ | $1.152 - 2.266i$ |
| $1.384 + 2.840i$ | $1.384 + 0.508i$ |
| $1.756 - 2.011i$ | $1.530 + 2.037i$ |
| $1.831 - 0.095i$ | $1.831 - 0.095i$ |
| $1.907 + 2.521i$ | $1.907 + 2.521i$ |
| $1.938 - 2.402i$ | $1.938 - 2.402i$ |
| $2.011 + 0.738i$ | $2.031 + 2.934i$ |
| $2.184 - 1.327i$ | $2.097 + 2.938i$ |
| $2.230 - 1.770i$ | $2.183 - 1.425i$ |
| $2.233 - 1.893i$ | $2.233 - 1.893i$ |

along a genus 2 surface without punctures and use the $\mathbb{Z}_2$ hyperelliptic involution. Every genus 2 conformal block is invariant under the corresponding $\mathbb{Z}_2$ symmetry acting by a rotation around the $x$ axis as follows:[8]

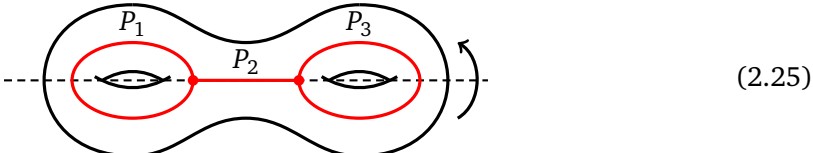

(2.25)

Thus any partition function of Virasoro TQFT on a hyperbolic three-manifold with only a genus 2 boundary must have the same property. In particular, we can produce two hyperbolic three-manifolds by cutting along a genus 2 surface and applying such a rotation. This yields in general non-equivalent hyperbolic three-manifolds, but with the same value of $Z_{\text{Vir}}$. Via the volume conjecture (2.1), this implies again that such a pair of manifolds has the same hyperbolic volume and the same value of the infinite product appearing as the one-loop determinant in (2.1).

As a concrete example that is perhaps more familiar and directly relevant to holography, consider the Euclidean wormhole of the form $\Sigma_2 \times [0, 1]$. Since the genus 2 surface is hyperelliptic, we can perform the hyperelliptic involution on one side, which formally leads to a different manifold, but with identical partition function. As explained in [1], the Virasoro TQFT partition function on the Euclidean wormhole is simply given by the partition function of Liouville CFT $Z_{\text{Liouville}}(\Sigma_{2,0}|\mathbf{m}_1, \mathbf{m}_2)$, where the left-moving moduli $\mathbf{m}_1$ are associated to the left boundary and the right-moving moduli $\mathbf{m}_2$ to the right-moving boundary. However, the Liouville partition function is already invariant when we apply the hyperelliptic involution to only $\mathbf{m}_1$, which means that the partition function of this twisted wormhole also equals the Liouville partition function.

## 2.5 Consistency conditions on the crossing kernels

The crossing kernels $\mathbb{F}$ and $\mathbb{S}$ on the four-punctured sphere and the once-punctured torus, respectively, are subject to a number of constraints known as the Moore-Seiberg consistency

---

[8]This would fail at genus 3 since we have in general two different Liouville momenta on the bottom and top of the middle loop, which get exchanged by this operation. Correspondingly, not every genus 3 surface is hyperelliptic.

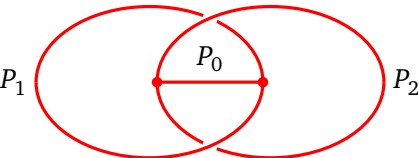

Figure 3: The network of Wilson lines used to derive the relation between the modular crossing kernel $\mathbb{S}$ and the sphere crossing kernel $\mathbb{F}$.

conditions [36]. We listed them in the Appendix of [1], see also [18]. They express consistency of the projective representation of the 2d mapping class group on the space of conformal blocks. For example, the modular crossing kernel $\mathbb{S}$ has to satisfy the $SL(2,\mathbb{Z})$ relations together with the Dehn twist $\mathbb{T}$ on the once-punctured torus.[9]

These relations are also necessary for the consistency of the three-dimensional theory. However, they can often be seen much easier from the three-dimensional perspective. We explain here one simple example that shows that $\mathbb{S}$ can be fully expressed in terms of $\mathbb{F}$ that we also use later in the paper. We should mention that this construction is standard in the context of modular tensor categories which can be viewed as the rational counterpart of Virasoro TQFT [37].[10]

Consider $S^3$ with a network of Wilson lines as in Figure 3. We recall that a juncture of Wilson lines was defined as follows in [1]:[11]

$$P_2 \underset{P_3}{\overset{P_1}{\diagup\!\!\!\diagdown}} \equiv \frac{1}{C_0(P_1,P_2,P_3)} \times P_2 \underset{P_3}{\overset{P_1}{\diagup\!\!\bigcirc\!\!\diagdown}} . \tag{2.26}$$

On the right hand side, we excise a spherical boundary around the puncture. The path integral then creates a state in the boundary Hilbert space, which is one-dimensional and hence can be canonically identified with $\mathbb{C}$ by fixing the standard normalization of the three-point function on the sphere.

The main point is now that the value of the partition function on the network of Wilson lines in Figure 3 can be computed in two different ways as follows.

Let us first consider the Heegaard splitting into two once-punctured tori. The two once-punctured tori are homeomorphic to tubular neighborhoods of the Wilson lines $P_1$ and $P_2$, respectively. The normalization of the juncture in (2.26) is chosen such that the Virasoro TQFT path integral on the once-punctured tori leads precisely to the respective conformal blocks on the boundary torus. The two once-punctured tori are interlocking and hence we have to apply an S-modular transformation. Being more careful about the definition of the S-modular transformation actually shows that we need the inverse of the modular crossing kernel $\mathbb{S}$. Since $\mathbb{S}$ squares to $e^{\pi i \Delta_0}$,[12] the inverse of $\mathbb{S}$ differs from $\mathbb{S}$ only by the phase $e^{-\pi i \Delta_0}$. In the end, we obtain for the partition function of the Wilson line network $M$,

$$Z_{\text{Vir}}(M) = e^{-\pi i \Delta_0} \int dP_1' \, \mathbb{S}_{P_1,P_1'}[P_0] \left\langle \overset{P_0}{\text{—•}} P_2 \middle| \overset{P_0}{\text{—•}} P_1' \right\rangle \tag{2.27}$$

$$= \frac{e^{-\pi i \Delta_0} \, \mathbb{S}_{P_1,P_2}[P_0]}{\rho_0(P_2) C_0(P_0,P_2,P_2)}, \tag{2.28}$$

---

[9]More precisely, the crossing kernels give rise to a projective representation of $SL(2,\mathbb{Z})$.

[10]We thank Sahand Seifnashri for explaining the MTC computation to us.

[11]$C_0(P_1,P_2,P_3)$ is the universal Liouville three-point function, see [1, eq. (2.17)].

[12]See [1, eq. (A.5a)].

where we applied [1, eq. (2.21)] for the evaluation of the inner product between conformal blocks. It reads

$$\langle \mathcal{F}_{g,n}^{\mathcal{C}}(\vec{P}) | \mathcal{F}_{g,n}^{\mathcal{C}}(\vec{P}') \rangle = \frac{\delta^{3g-3+n}(\vec{P}-\vec{P}')}{\prod_{\text{cuffs } a} \rho_0(P_a) \prod_{\text{pair of pants } (i,j,k)} C_0(P_i, P_j, P_k)} \ . \tag{2.29}$$

Here, $\mathcal{F}_{g,n}^{\mathcal{C}}$ are the genus-$g$ $n$-point Virasoro blocks in a particular OPE channel $\mathcal{C}$, $C_0(P_1, P_2, P_3)$ is the Liouville three-point function, and $\rho_0(P)$ the inverse of the two-point function. In other words, conformal blocks in a channel $\mathcal{C}$ are orthogonal with a density given by the inverse OPE density of Liouville theory. See [1] for our conventions for the Liouville structure constants.

We can alternatively compute the partition function by a Heegaard splitting along a four-punctured sphere containing the Wilson line $P_0$ and the stubs of the Wilson lines $P_1$ and $P_2$. We have in hopefully obvious notation

$$Z_{\text{Vir}}(M) = \int dP \ \mathbb{F}_{P_0,P} \begin{bmatrix} P_2 & P_1 \\ P_2 & P_1 \end{bmatrix} Z_{\text{Vir}}\left( \begin{array}{c} P_1 \ P \end{array} \ P_2 \right) \tag{2.30}$$

$$= \int dP \ \mathbb{F}_{P_0,P} \begin{bmatrix} P_2 & P_1 \\ P_2 & P_1 \end{bmatrix} e^{2\pi i (\Delta - \Delta_1 - \Delta_2)} Z_{\text{Vir}}\left( \begin{array}{c} P_1 \ P \end{array} \ P_2 \right) \tag{2.31}$$

$$= \int dP \ \mathbb{F}_{P_0,P} \begin{bmatrix} P_2 & P_1 \\ P_2 & P_1 \end{bmatrix} \frac{e^{2\pi i (\Delta - \Delta_1 - \Delta_2)}}{C_0(P, P_1, P_2)} \ , \tag{2.32}$$

where we used the braiding move twice in the second line. In the last line we recognize the Euclidean wormhole with two three-punctured sphere on both ends, which evaluates to the Liouville three-point function. Taking into account the normalization in eq. (2.26), we get an inverse structure constant.

Comparing (2.28) and (2.32) then expresses the modular crossing kernel fully in terms of the sphere crossing kernel,

$$\mathbb{S}_{P_1, P_2}[P_0] = \int dP \ \frac{\rho_0(P_2) C_0(P_0, P_2, P_2)}{C_0(P, P_1, P_2)} e^{\pi i (2\Delta + \Delta_0 - 2\Delta_1 - 2\Delta_2)} \mathbb{F}_{P_0,P} \begin{bmatrix} P_2 & P_1 \\ P_2 & P_1 \end{bmatrix} . \tag{2.33}$$

Using the explicit form of the sphere crossing kernel given e.g. in eq. (2.42a) of [1], one can use various known identities of the involved integrals of special functions to derive the known expression of the modular fusion kernel from this integral formula [18, 26].

This identity can also be derived from a two-dimensional point of view by requiring the consistency of the representation of the mapping class group on the space of conformal blocks on the two-punctured torus. It is in fact a special case of the corresponding Moore-Seiberg relation. However, the corresponding derivation is much more complicated and subtle than the three-dimensional point of view.

## 3 Holographic examples

We now move on to holographic applications of the Virasoro TQFT formalism. We will mostly focus on multi-boundary wormholes that have direct implications for the description of the holographic dual of 3d gravity in terms of an ensemble of CFT data. In order to set the stage for this discussion, let us briefly recapitulate the ensemble description of AdS$_3$ gravity.

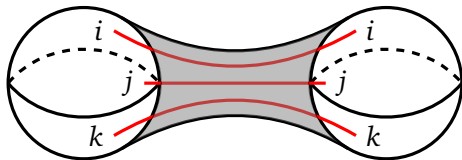

Figure 4: The Euclidean wormhole with the topology of a three-punctured sphere times an interval that contributes to the variance of the structure constants in the ensemble description of the dual of AdS$_3$ gravity.

In [10] it was shown that averaged products of CFT observables in a Gaussian ensemble for the CFT data defined by

$$\overline{c_{ijk}} = 0 \,, \tag{3.1a}$$

$$\overline{c_{ijk}c^*_{\ell mn}} = C_0(P_i, P_j, P_k)C_0(\bar{P}_i, \bar{P}_j, \bar{P}_k)\left(\delta_{i\ell}\delta_{jm}\delta_{kn} + (-1)^{\ell_i+\ell_j+\ell_k}\delta_{i\ell}\delta_{jn}\delta_{km} + 4\text{ permutations}\right) \,, \tag{3.1b}$$

together with a Cardy spectrum of heavy states, agree with the on-shell actions (and in certain cases, the one-loop determinants) of suitable Euclidean wormholes in semiclassical AdS$_3$ gravity coupled to massive point particles. Here $\ell_i = P_i^2 - \bar{P}_i^2$ is the spin of the corresponding primary. The averaged CFT quantities are computed by performing a simultaneous conformal block decomposition of the observables and computing Wick contractions of the structure constants using (3.1). The gravity computations were mostly restricted to two-boundary wormholes with topology $\Sigma \times [0, 1]$, with $\Sigma$ a (possibly punctured) Riemann surface, corresponding to two-copy averaged observables on the CFT ensemble side. Indeed one may view (3.1) as being determined by an explicit computation of the 3d gravity partition function on a Euclidean wormhole with the topology of a three-punctured sphere times an interval [10]; see figure 4.

In [1] the correspondence between two-boundary Euclidean wormhole partition functions and averaged products of CFT observables was extended to finite central charge using Virasoro TQFT. In particular the TQFT partition function on the Euclidean wormhole was computed, with the result

$$Z_{\text{Vir}}(\Sigma \times [0, 1]|\mathbf{m}_1, \mathbf{m}_2) = Z_{\text{Liouville}}(\Sigma|\mathbf{m}_1, \mathbf{m}_2) \,, \tag{3.2}$$

where $\mathbf{m}_1, \mathbf{m}_2$ collectively denote the moduli of the Riemann surfaces at the two boundaries, and $Z_{\text{Liouville}}(\Sigma)$ is the correlation function on $\Sigma$ in Liouville CFT. $|Z_{\text{Vir}}(\Sigma \times [0, 1])|^2$ agrees with the the averaged CFT computations performed in the Gaussian ensemble (3.1), and its large-$c$ expansion agrees with the semiclassical gravity saddle-point computations in [10].

Except in certain very special cases it is not clear how to compute the gravity path integral on configurations with more than two asymptotic boundaries in the metric formalism. Such configurations in particular encode non-Gaussian corrections to the ensemble formulation of the boundary theory defined in (3.1), which are known to be needed for the internal consistency of the ensemble description from a variety of points of view [11, 21, 38]. For example, the existence of a Gaussian contraction often depends on the specific choice of channel in the conformal block decomposition of the CFT observables; crossing symmetry then requires non-Gaussian statistics in the dual channel in order to reproduce the result in the channel where the Gaussian contraction exists. Hence non-Gaussian corrections, which are necessary for an internally consistent description of the boundary ensemble, are not presently accessible in the metric formulation of AdS$_3$ gravity.

In the remainder of this section we will study Euclidean wormholes in AdS$_3$ gravity with more than two asymptotic boundaries using Virasoro TQFT. We will mostly focus on wormholes with more than two three-punctured sphere boundaries, since these determine the leading contributions to higher moments of the structure constants in the ensemble description of the

holographic dual. We will see in some examples that the resulting non-Gaussian statistics precisely affirm the consistency of the results computed in the Gaussian ensemble.

There may also be non-Gaussian corrections to (3.1) associated to Euclidean wormholes with two three-punctured sphere boundaries but with higher topology in the bulk. We will not study such corrections here, but let us briefly mention that we have already encountered such a correction associated with a higher-topology wormhole. In section 2.5 we studied a configuration of Wilson lines equivalent to the following two-boundary wormhole with linked Wilson lines

$$M = \quad . \tag{3.3}$$

The TQFT partition function on this wormhole may be computed as described in section 2.5. One finds the following for the wormhole partition function

$$Z_{\text{Vir}}(M) = \frac{C_0(P_i, P_j, P_j) \mathbb{S}_{P_j P_k}[P_i]}{\rho_0(P_k)}, \tag{3.4}$$

corresponding to the following averaged product of structure constants that would otherwise vanish (in the case that $j \neq k$) in the Gaussian ensemble[13]

$$\overline{c_{ijj} c_{ikk}^*} = |Z_{\text{Vir}}(M)|^2 = \left| \frac{C_0(P_i, P_j, P_j) \mathbb{S}_{P_j P_k}[P_i]}{\rho_0(P_k)} \right|^2. \tag{3.5}$$

## 3.1 Cyclic defect wormholes

A simple class of examples that demonstrate the practical utility of the TQFT reformulation of 3d gravity is provided by multi-boundary wormholes with defects connecting the sphere boundaries. For concreteness, consider the case where each boundary is a four-punctured sphere with defects connected in a pairwise cyclic way. See figure 5 for a depiction of such a wormhole with three boundaries. In [10] it was argued that such on-shell wormholes contribute to the following averaged product of four-point functions

$$\overline{\langle \mathcal{O}_1 \mathcal{O}_2 \mathcal{O}_3 \mathcal{O}_4 \rangle \langle \mathcal{O}_3 \mathcal{O}_4 \mathcal{O}_5 \mathcal{O}_6 \rangle \cdots \langle \mathcal{O}_{2k-1} \mathcal{O}_{2k} \mathcal{O}_1 \mathcal{O}_2 \rangle} \tag{3.6}$$

in the ensemble of CFT data dual to semiclassical 3d gravity.[14] The Gaussian ensemble hence makes a specific prediction for the gravitational partition function of these wormholes [10]:

$$Z_{\text{grav}}(M_k) \overset{?}{=} \left| \int_0^\infty \mathrm{d}P \, \rho_0(P) C_0(P_1, P_2, P) C_0(P_3, P_4, P) \cdots C_0(P_{2k-1}, P_{2k}, P) \right.$$

$$\left. \times \quad (z_1) \quad (z_2) \cdots \quad (z_k) \right|^2. \tag{3.7}$$

Here we have used the notation $M_k$ to refer to the $k$-boundary sphere four-point wormhole, the stick diagrams are shorthand for the conformal blocks as usual, and $z_i$ refers to the cross-ratio of the defect insertions on the $i^{\text{th}}$ boundary. The effect of the Gaussian average is to set all the internal weights equal in this particular conformal block decomposition of the wormhole partition function.

---

[13]Strictly speaking, we get a different bulk manifold when we exchange the two ends of the Wilson line labelled by $k$. Exchanging them leads to a braiding phase $e^{\pi i \Delta_j}$. Summing over both possibilities imposes that the spin of $i$ has to be even.

[14]Wormholes of this sort also contribute to the Renyi entropies of certain coarse-grained states in 2d CFT [39].

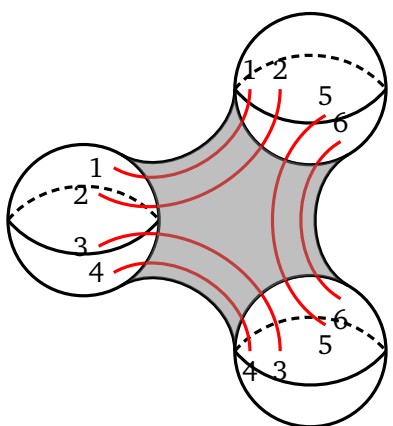

Figure 5: The three-boundary sphere four-point wormhole $M_3$.

It is not at all clear how to compute the wormhole partition function of the $k$-boundary sphere four-point wormhole in the metric formalism of 3d gravity, even in the semiclassical limit. Here we will describe how this wormhole partition function may be straightforwardly computed in the Virasoro TQFT, reproducing the expectation from the Gaussian ensemble (3.7).

For concreteness and brevity of the equations we consider here the case $k = 3$, but emphasize that the generalization to higher $k$ is completely straightforward. The idea is to view the wormhole as a compression body as indicated in figure 6. To compute the partition function on the corresponding compression body we insert a complete set of states in the Hilbert space of the inner boundaries. This produces a particular state in the Hilbert space of the outer boundary. Proceeding in this way we have

$$Z_{\text{Vir}}(M_3) = \int_0^\infty dP_a\, dP_b\, \rho_0(P_a)\rho_0(P_b) C_0(P_1, P_2, P_a) C_0(P_3, P_4, P_a) C_0(P_3, P_4, P_b)$$

$$\times C_0(P_5, P_6, P_b) \qquad \text{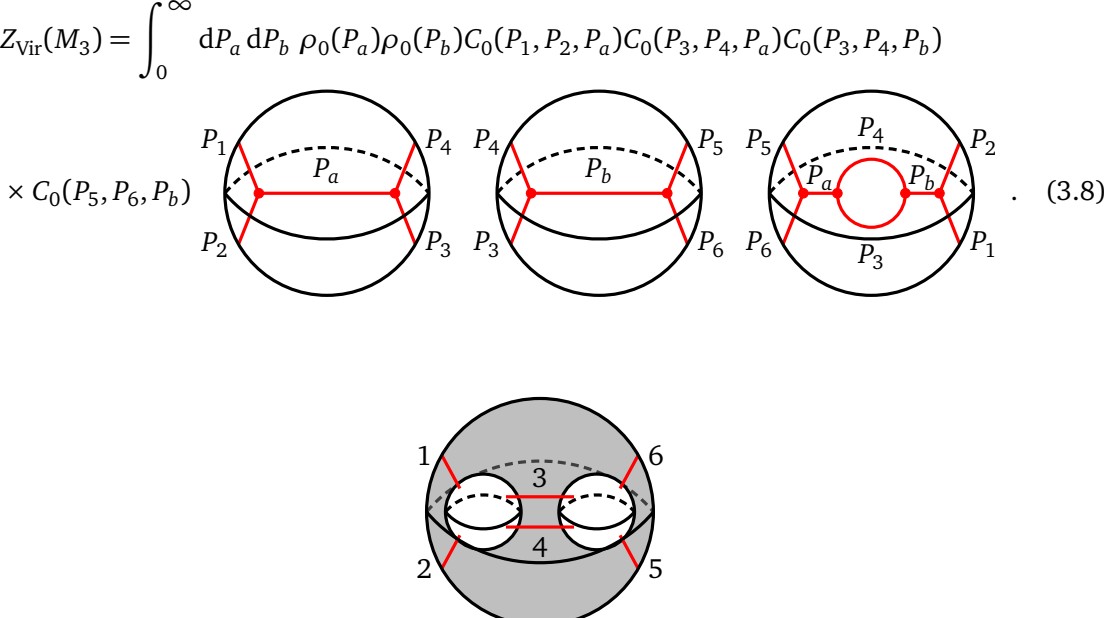} \qquad . \quad (3.8)$$

Figure 6: The three-boundary sphere four-point wormhole as a compression body.

We have temporarily restored the sphere boundaries in representing the conformal blocks in order to emphasize that the last conformal block should be interpreted as a state in the Hilbert space of the outer sphere boundary with a loop of Wilson lines in the interior, *not* as a higher-genus conformal block. In particular we can remove the loop by recalling the TQFT identity [1]

$$
\overset{P_4}{\underset{P_3}{P_a \bigcirc P_b}} = \frac{\delta(P_a - P_b)}{\rho_0(P_a) C_0(P_3, P_4, P_a)} \quad \underset{}{P_a} \quad , \tag{3.9}
$$

which leads us to

$$
Z_{\mathrm{Vir}}(M_3) = \int_0^\infty \mathrm{d}P_a \, \rho_0(P_a) C_0(P_1, P_2, P_a) C_0(P_3, P_4, P_a) C_0(P_5, P_6, P_a)
$$

$$
\times \; \overset{P_1}{\underset{P_2}{\searrow}} \overset{P_a}{\underset{}{\longrightarrow}} \overset{P_4}{\underset{P_3}{\swarrow}} \quad \overset{P_4}{\underset{P_3}{\searrow}} \overset{P_a}{\underset{}{\longrightarrow}} \overset{P_5}{\underset{P_6}{\swarrow}} \quad \overset{P_5}{\underset{P_6}{\searrow}} \overset{P_a}{\underset{}{\longrightarrow}} \overset{P_2}{\underset{P_1}{\swarrow}} \; . \tag{3.10}
$$

The generalization to the case of $k$ four-punctured sphere boundaries follows immediately by viewing $M_k$ as a compression body with $k-1$ inner boundaries and repeated application of the identity (3.9). Upon squaring the TQFT partition function to obtain the 3d gravity partition function, we hence verify (3.7), the prediction from the averaged product of $k$ sphere four-point functions in the Gaussian ensemble. Much like the case of the two-boundary Euclidean wormhole revisited in [1], the correspondence between the averaged CFT quantities and the gravity partition function on a fixed topology persists beyond the semiclassical limit.

## 3.2 Four-boundary non-Gaussianity wormhole

Consider a wormhole with four three-punctured spheres as asymptotic boundaries, with defects threading the bulk of the wormhole in the following tetrahedral configuration

$$
M = \tag{3.11}
$$

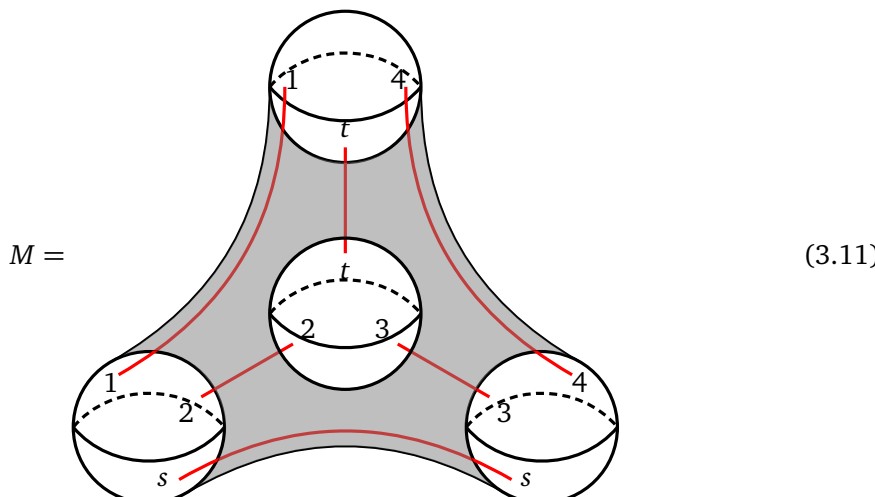

The gravity path integral on this wormhole should compute the following connected part of the fourth moment of structure constants in the dual description of 3d gravity in terms of an ensemble of CFT data

$$
|Z_{\mathrm{Vir}}(M)|^2 \leftrightarrow \overline{c_{12s} c_{s34} c_{14t} c_{t32}} \, . \tag{3.12}
$$

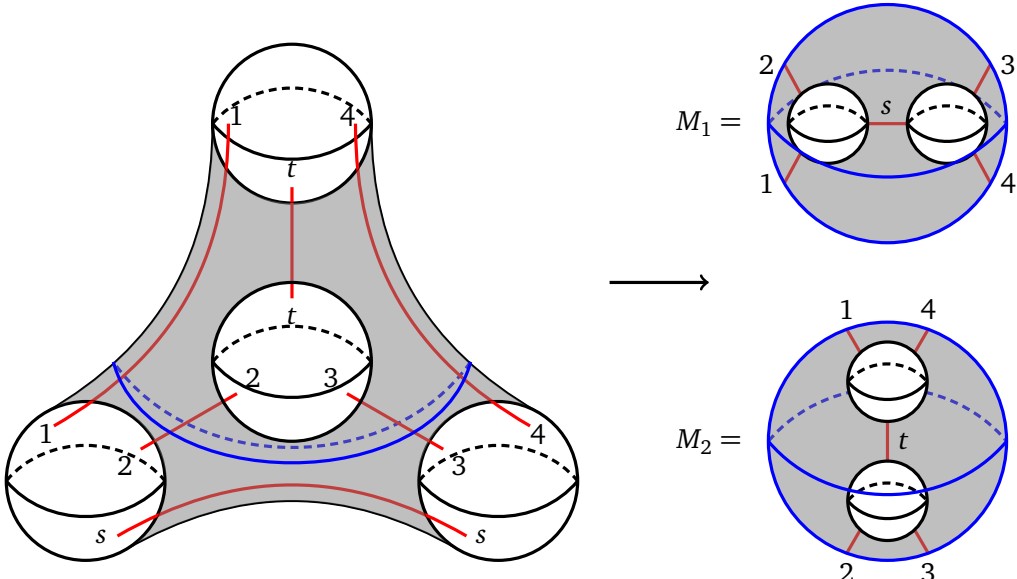

Figure 7: The Heegaard splitting of the four-boundary wormhole $M$ into two generalized compression bodies $M_1$ and $M_2$, each with the topology of a three-ball with two three-balls drilled out in its interior and with Wilson lines connecting the two-sphere boundaries as shown in the figure. The TQFT path integral on each of $M_1$ and $M_2$ prepares a state in the Hilbert space of the four-punctured sphere.

**Computation via Heegaard splitting.** It straightforward to apply the Heegaard splitting technique described in detail in [1] to compute the Virasoro TQFT partition function on the four-boundary wormhole. For instance, we can cut $M$ along a four-punctured sphere through the bulk of the wormhole as pictured in figure 7. This cuts the four-boundary wormhole into two generalized compression bodies $M_1$ and $M_2$. Each compression body has an outer boundary given by a four-punctured sphere and two three-punctured sphere inner boundaries. The Virasoro TQFT path integral on each compression body prepares a state in the Hilbert space of the four-punctured sphere, and the inner product of these states computes the TQFT partition function on the four-boundary wormhole. Using (2.26) to write the three-punctured sphere boundaries in terms of trivalent Wilson line junctions, the TQFT partition functions on the compression bodies are given by

$$\langle Z_{\mathrm{Vir}}(M_1)| = C_0(P_1, P_2, P_s) C_0(P_3, P_4, P_s) \left\langle \begin{array}{c} \end{array} \right| , \tag{3.13a}$$

$$|Z_{\mathrm{Vir}}(M_2)\rangle = C_0(P_1, P_4, P_t) C_0(P_2, P_3, P_t) \left| \begin{array}{c} \end{array} \right\rangle . \tag{3.13b}$$

Up to the $C_0$ factors, the compression body partition functions are given by individual sphere four-point conformal blocks in the $s$- and the $t$-channel. The inner product of these states is proportional to the Virasoro fusion kernel essentially by definition:

$$\left\langle \begin{array}{c} P_1 \quad P_s \quad P_4 \\ P_2 \qquad P_3 \end{array} \middle| \begin{array}{c} P_1 \quad P_4 \\ P_t \\ P_2 \quad P_3 \end{array} \right\rangle$$

$$= \int dP_s'\, \mathbb{F}_{P_t P_s'}\begin{bmatrix} P_1 & P_2 \\ P_4 & P_3 \end{bmatrix}\left\langle \begin{array}{c} P_1 \quad P_s \quad P_4 \\ P_2 \qquad P_3 \end{array} \middle| \begin{array}{c} P_1 \quad P_s' \quad P_4 \\ P_2 \qquad P_3 \end{array} \right\rangle \tag{3.14}$$

$$= \frac{\mathbb{F}_{P_t P_s}\begin{bmatrix} P_1 & P_2 \\ P_4 & P_3 \end{bmatrix}}{\rho_0(P_s)C_0(P_1,P_2,P_s)C_0(P_3,P_4,P_s)} \tag{3.15}$$

$$= \frac{\mathbb{F}_{P_s P_t}\begin{bmatrix} P_1 & P_4 \\ P_2 & P_3 \end{bmatrix}}{\rho_0(P_t)C_0(P_1,P_4,P_t)C_0(P_2,P_3,P_t)}. \tag{3.16}$$

In the penultimate line we computed the inner product by expanding the $t$-channel block in a complete basis of $s$-channel blocks using the Ponsot-Teschner fusion kernel [15,16], and in the last line we did the reverse. The equivalence of these two expressions is not a priori obvious without appealing to consistency of the conformal block inner product, but it is guaranteed by for example a special case of the pentagon identity, which is one of the Moore-Seiberg consistency conditions satisfied by the fusion kernel. In fact, this combination has a tetrahedral symmetry inherited from the bulk Wilson line configuration that is obscured by this presentation. Indeed, it can be rewritten in a manifestly tetrahedrally symmetric form in terms of the Virasoro $6j$ symbol in the Racah-Wigner normalization [17] as follows

$$\left\langle \begin{array}{c} P_1 \quad P_s \quad P_4 \\ P_2 \qquad P_3 \end{array} \middle| \begin{array}{c} P_1 \quad P_4 \\ P_t \\ P_2 \quad P_3 \end{array} \right\rangle$$

$$= \frac{\begin{Bmatrix} P_1 & P_2 & P_s \\ P_3 & P_4 & P_t \end{Bmatrix}}{\sqrt{C_0(P_1,P_2,P_s)C_0(P_3,P_4,P_s)C_0(P_1,P_4,P_t)C_0(P_2,P_3,P_t)}}. \tag{3.17}$$

The upshot is that the Virasoro TQFT partition function on the four-boundary wormhole can be expressed in terms of the Virasoro $6j$ symbol via the following inner product in the Hilbert space of the four-punctured sphere

$$Z_{\text{Vir}}(M) = \langle Z_{\text{Vir}}(M_1) | Z_{\text{Vir}}(M_2) \rangle \tag{3.18}$$

$$= C_0(P_1,P_2,P_s)C_0(P_3,P_4,P_s)C_0(P_1,P_4,P_t)C_0(P_2,P_3,P_t)$$

$$\times \left\langle \begin{array}{c} P_1 \quad P_s \quad P_4 \\ P_2 \qquad P_3 \end{array} \middle| \begin{array}{c} P_1 \quad P_4 \\ P_t \\ P_2 \quad P_3 \end{array} \right\rangle \tag{3.19}$$

$$= \sqrt{C_0(P_1,P_2,P_s)C_0(P_3,P_4,P_s)C_0(P_1,P_4,P_t)C_0(P_2,P_3,P_t)}\begin{Bmatrix} P_1 & P_2 & P_s \\ P_3 & P_4 & P_t \end{Bmatrix}. \tag{3.20}$$

**Consistency with boundary ensemble description.** The gravity partition function $Z_{\mathrm{grav}}(M) = |Z_{\mathrm{Vir}}(M)|^2$ on the four-boundary wormhole (3.20) makes a concrete prediction for the connected contribution to the fourth moment of structure constants in the description of 3d gravity in terms of an ensemble of CFT data:[15]

$$\overline{c_{12s}c_{s34}c_{14t}c_{t32}} \supset |Z_{\mathrm{Vir}}(M)|^2 = \sqrt{\overline{c_{12s}^2}\,\overline{c_{s34}^2}\,\overline{c_{14t}^2}\,\overline{c_{t32}^2}} \left| \begin{Bmatrix} P_1 & P_2 & P_s \\ P_3 & P_4 & P_t \end{Bmatrix} \right|^2 . \tag{3.21}$$

This represents the leading correction to the Gaussian ensemble elucidated in [10]. We say that the fourth moment *contains* this contribution (rather than being literally equal to it) because there may be corrections to (3.21) associated with wormholes with the same boundaries but with higher topology in the bulk. It is expected that in the semiclassical limit such contributions are parametrically suppressed and hence that (3.21) represents the leading contribution to the fourth moment.

Here we will see that this non-Gaussian correction in fact exactly ensures the internal consistency of the results predicted by the Gaussian ensemble.

To illustrate the point, consider the two-boundary Euclidean wormhole with the topology of a (possibly punctured) Riemann surface $\Sigma$ times an interval. The gravity path integral on the Euclidean wormhole is given by the square of (3.2), the corresponding observable in Liouville CFT with the moduli on the two sides paired. This agrees with the averaged product of CFT observables in the Gaussian ensemble (3.1). However the computation in the Gaussian ensemble often relies on the choice of a specific channel in the conformal block decomposition; this is obviously inconsistent with crossing symmetry of the ensemble. Associativity of the OPE then requires non-Gaussian statistics in order to reproduce this result in other channels. Relatedly, while the Gaussian ensemble is crossing symmetric on average, higher moments of the crossing equation do not vanish; this has recently been emphasized in [11].

For concreteness, consider in particular the averaged product of four-point functions of local operators $\mathcal{O}_i$. In the Gaussian ensemble, we have

$$\overline{\langle \mathcal{O}_1(0)\mathcal{O}_2(z,\bar{z})\mathcal{O}_3(1)\mathcal{O}_4(\infty)\rangle \langle \mathcal{O}_1(0)\mathcal{O}_2(z',\bar{z}')\mathcal{O}_3(1)\mathcal{O}_4(\infty)\rangle^*}$$

$$= \sum_{s,s'} \overline{c_{12s}c_{34s}c_{12s'}^* c_{34s'}^*} \left| \raisebox{-1em}{\begin{array}{c}\text{[s-channel block diagram }(z)\text{ and }(z')]\end{array}} \right|^2 \tag{3.22}$$

$$= \left| \int dP_s\, \rho_0(P_s) C_0(P_1,P_2,P_s) C_0(P_3,P_4,P_s) \raisebox{-1em}{\begin{array}{c}\text{[s-channel block diagram }(z)\text{ and }(z')]\end{array}} \right|^2 \tag{3.23}$$

$$= \left| Z_{\mathrm{Liouville}}(P_1,P_2,P_3,P_4|z,z') \right|^2 . \tag{3.24}$$

Here we expanded the four-point functions in the same OPE channel, and performed the Gaussian contractions in the third line using eq. (3.1). If we had instead expanded one four-point function in the S-channel and the other in the T-channel, we would have gotten zero in the Gaussian ensemble since

$$\overline{c_{12s}c_{s34}c_{14t}c_{t32}}\Big|_{\mathrm{Gaussian}} = 0 , \tag{3.25}$$

for distinct external operators.[16] This is obviously inconsistent with basic principles of conformal field theory. The result for the averaged product of four-point functions in terms of the four-point function in Liouville CFT is equal to the partition function of 3d gravity coupled

---

[15]This fourth moment has previously appeared in [11] where it was argued for by requiring that the variance of the crossing equation vanish, and in [38] where it followed from genus-three modular invariance (using similar logic as that which shows that the variance should be given by the $C_0$ formula).

[16]We thank Vladimir Narovlansky for asking a question that raised this point.

to point particles on the Euclidean wormhole with the topology of a four-punctured sphere times an interval, so we seek a correction to the Gaussian ensemble that preserves (3.24). If we supplement the Gaussian ensemble with the fourth moment (3.21) as computed by the four-boundary wormhole, we instead have

$$\overline{\langle\mathcal{O}_1(0)\mathcal{O}_2(z,\bar{z})\mathcal{O}_3(1)\mathcal{O}_4(\infty)\rangle\langle\mathcal{O}_1(0)\mathcal{O}_2(z',\bar{z}')\mathcal{O}_3(1)\mathcal{O}_4(\infty)\rangle^*}$$

$$= \sum_{s,t}\overline{c_{12s}c_{s34}c_{41t}^*c_{t23}^*}\left|\ \text{(diagram)}\ (z)\ \text{(diagram)}\ (z')\right|^2 \qquad (3.26)$$

$$= \left|\int \mathrm{d}P_s\,\mathrm{d}P_t\,\rho_0(P_s)C_0(P_1,P_2,P_s)C_0(P_3,P_4,P_s)\mathbb{F}_{P_sP_t}\begin{bmatrix}P_1 & P_4\\ P_2 & P_3\end{bmatrix}\right.$$

$$\times \left.\ \text{(diagram)}\ (z)\ \text{(diagram)}\ (z')\right|^2 \qquad (3.27)$$

$$= \left|Z_{\text{Liouville}}(P_1,P_2,P_3,P_4|z,z')\right|^2\,, \qquad (3.28)$$

in agreement with the previous computation and with the wormhole partition function.

**On braiding and the $u$-channel.**   In the discussion so far we have suppressed an important subtlety. In 2d CFT, the structure constants are not strictly invariant under permutations of the three operators. For example, swapping a pair of operators leads to a sign that depends on the sum of the spins of the three operators

$$c_{ikj} = (-1)^{\ell_i+\ell_j+\ell_k}c_{ijk}\,. \qquad (3.29)$$

This is inherited from reality properties of the structure constants: they are real if the sum of spins is even and imaginary if the sum of spins is odd, and the swap complex-conjugates the structure constants, $c_{ikj} = c_{ijk}^*$. Similarly, in the computation of wormhole partition functions with bulk Wilson lines via Heegaard splitting, there may be crossings of lines that need to be undone via braiding operations. These braidings introduce phases that depend on the conformal weights.

In general, we can read off the ordering of the structure constants from a bulk manifold by fixing a cyclic ordering and reading the labels around three-punctured boundaries cyclically. The same applies in CFT computations, where we read off the labels of the structure constants cyclically around every vertex in the conformal blocks.[17]

---

[17]The overall cyclic direction does not matter since every label appears twice and thus cancels if we reverse the overall cyclic direction.

As a simple example of a wormhole computation for which such braidings are essential, consider the following four-boundary wormhole:

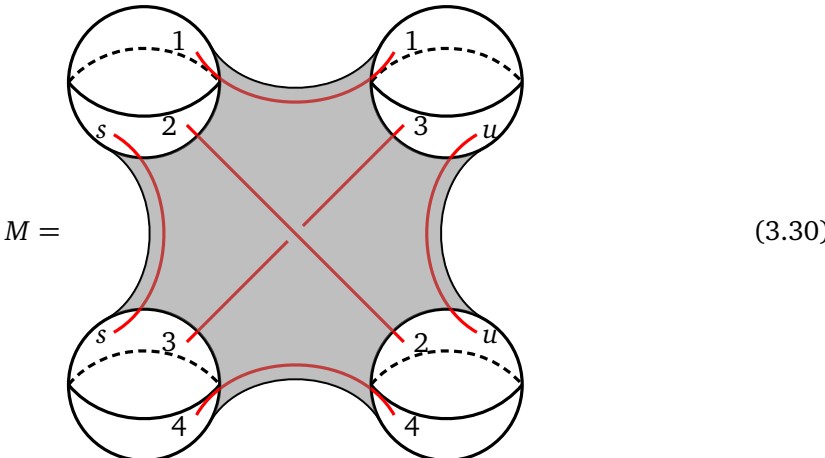

$$M = \qquad\qquad\qquad\qquad\qquad\qquad\qquad\qquad\qquad (3.30)$$

which is essentially the same as (3.11). The boundaries of this wormhole are three-punctured spheres corresponding to the structure constants that appear in the $s$- and $u$-channel conformal block decompositions of the sphere four-point function $\langle \mathcal{O}_1 \mathcal{O}_2 \mathcal{O}_3 \mathcal{O}_4 \rangle$.

We compute the TQFT partition function as before by splitting along a four-punctured sphere in the bulk. Undoing the crossing of the Wilson lines and computing the inner product in the Hilbert space of the splitting surface leads to the following result for the TQFT partition function on this four-boundary wormhole

$$Z_{\text{Vir}}(M) = \sqrt{\mathsf{C}_{12s}\mathsf{C}_{s34}\mathsf{C}_{31u}\mathsf{C}_{u42}} \begin{Bmatrix} P_1 & P_2 & P_s \\ P_4 & P_3 & P_u \end{Bmatrix} e^{\pi i (P_1^2 + P_4^2 - P_s^2 - P_u^2)}. \qquad (3.31)$$

Here we have introduced the shorthand

$$\mathsf{C}_{ijk} \equiv C_0(P_i, P_j, P_k), \qquad (3.32)$$

We notice the presence of an additional phase compared to (3.20). This result follows from taking the inner product between an $s$- and a $u$-channel Virasoro conformal block, and hence this phase may be understood in terms of the crossing transformation that relates $s$- and $u$-channel blocks. This crossing transformation is given by

$$\overset{2\quad 3}{\underset{s}{\underset{1\rule{0pt}{0pt}\qquad\quad 4}{\bigsqcup}}} = \int_0^\infty dP_u\, e^{\pi i (P_1^2 + P_4^2 - P_s^2 - P_u^2)}\, \mathbb{F}_{P_s P_u}\begin{bmatrix} P_1 & P_3 \\ P_2 & P_4 \end{bmatrix} \overset{2\quad 3}{\underset{u}{\underset{1\qquad\quad 4}{\times}}}. \qquad (3.33)$$

The combination that appears on the right-hand side is sometimes referred to as the "R-matrix." The semiclassical near-extremal limit of the R-matrix governs the out-of-time-order four-point function in the Schwarzian theory [19].

Hence for the following fourth moment of CFT structure constants we find

$$\overline{c_{12s}c_{s34}c_{31u}c_{u42}} \supset |Z_{\text{Vir}}(M)|^2 = (-1)^{\ell_1+\ell_4+\ell_s+\ell_u} \sqrt{\overline{c_{12s}^2}\,\overline{c_{s34}^2}\,\overline{c_{13u}^2}\,\overline{c_{u24}^2}} \left| \begin{Bmatrix} P_1 & P_2 & P_s \\ P_4 & P_3 & P_u \end{Bmatrix} \right|^2. \qquad (3.34)$$

This is exactly consistent with the previous result (3.21) upon relabeling $t \to u$, $4 \leftrightarrow 3$ and making use of the exchange property (3.29). It is also consistent with the averaged product of sphere four-point functions in the Gaussian ensemble, where we expand one four-point function in the $s$-channel and the other in the $u$-channel.

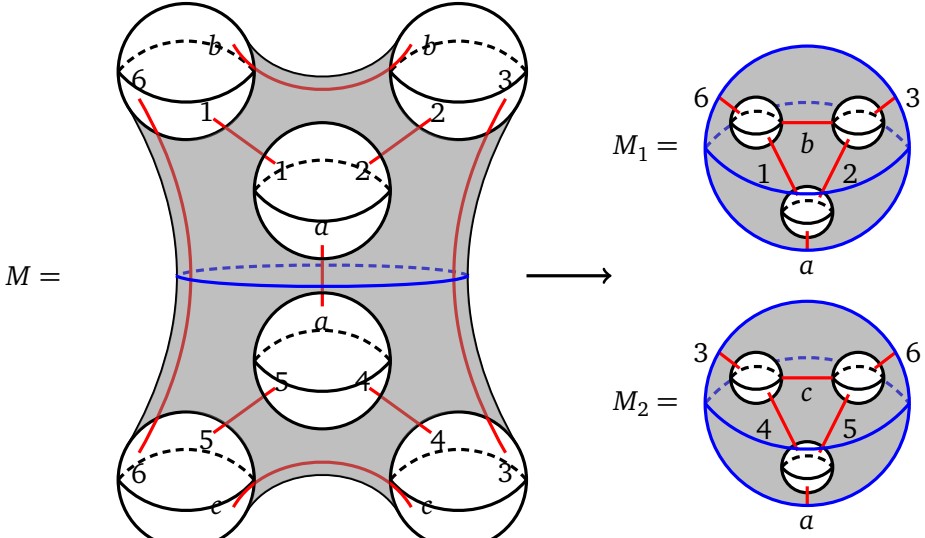

Figure 8: A Heegaard splitting of the wormhole with six three-punctured sphere boundaries. Each constituent compression body is equivalent to the four-boundary wormhole studied in section 3.2.

## 3.3 Many-boundary wormholes and higher non-Gaussianities

### 3.3.1 A simple six-boundary example

Consider the following wormhole with six three-punctured sphere boundaries

$$M = \qquad\qquad . \qquad\qquad (3.35)$$

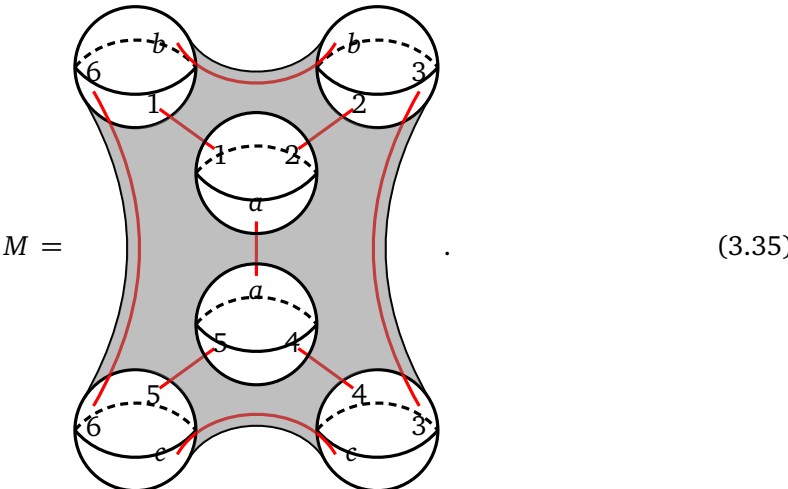

As indicated by the diagram, it contributes to the following sixth moment of CFT structure constants

$$|Z_{\text{Vir}}(M)|^2 \longleftrightarrow \overline{c_{12a}c_{2b3}c_{3c4}c_{45a}c_{5c6}c_{6b1}} \,. \qquad (3.36)$$

There are several Heegaard splittings that one could employ to compute the Virasoro TQFT partition function on this wormhole, but the simplest is indicated in figure 8: we cut the wormhole through the bulk along a three-punctured sphere. This divides $M$ into two generalized compression bodies $M_1$ and $M_2$, each of which is itself a four-boundary wormhole of the type described in the previous subsection. The partition function of Virasoro TQFT on the

generalized compression bodies was computed in (3.20) as

$$|Z_{\text{Vir}}(M_1)\rangle = \sqrt{C_{12a}C_{2b3}C_{6b1}C_{a36}} \begin{Bmatrix} P_1 & P_2 & P_a \\ P_3 & P_6 & P_b \end{Bmatrix},$$ (3.37a)

$$|Z_{\text{Vir}}(M_2)\rangle = \sqrt{C_{3c4}C_{45a}C_{5c6}C_{a36}} \begin{Bmatrix} P_3 & P_4 & P_c \\ P_5 & P_6 & P_a \end{Bmatrix}.$$ (3.37b)

Then the Virasoro TQFT partition function on the six-boundary wormhole is given by the following inner product between these states in the Hilbert space of the shared three-punctured sphere boundary

$$Z_{\text{Vir}}(M) = \langle Z_{\text{Vir}}(M_1)|Z_{\text{Vir}}(M_2)\rangle$$ (3.38)

$$= \sqrt{C_{12a}C_{2b3}C_{3c4}C_{45a}C_{5c6}C_{6b1}} \begin{Bmatrix} P_1 & P_2 & P_a \\ P_3 & P_6 & P_b \end{Bmatrix} \begin{Bmatrix} P_3 & P_4 & P_c \\ P_5 & P_6 & P_a \end{Bmatrix}.$$ (3.39)

Notice that here the only effect of the inner product is to divide by the extra factor of $C_0(P_3, P_a, P_6)$.

This particular Heegaard splitting of the six-boundary wormhole is far from unique: for example, we could have cut it through a five-punctured sphere, or along three four-punctured spheres. In all cases, the corresponding splittings yield the same result (3.39) for the TQFT partition function.

This wormhole partition function implies that the corresponding sixth moment of CFT structure constants is given by

$$\overline{c_{12a}c_{2b3}c_{3c4}c_{45a}c_{5c6}c_{6b1}} \supset \sqrt{\overline{c_{12a}^2}\,\overline{c_{2b3}^2}\,\overline{c_{3c4}^2}\,\overline{c_{45a}^2}\,\overline{c_{5c6}^2}\,\overline{c_{6b1}^2}} \left| \begin{Bmatrix} P_1 & P_2 & P_a \\ P_3 & P_6 & P_b \end{Bmatrix} \begin{Bmatrix} P_3 & P_4 & P_c \\ P_5 & P_6 & P_a \end{Bmatrix} \right|^2.$$ (3.40)

**Consistency with boundary ensemble description.** Much like the fourth moment of the structure constants inferred from the four-boundary wormhole of section 3.2, the sixth moment (3.40) is needed for consistency of the description of the boundary theory in terms of an ensemble of CFT data. There are a variety of ways to see this. Roughly, for each Heegaard splitting of the wormhole, there is a corresponding product of CFT observables for which consistency of the ensemble description requires that the appropriate moment of CFT data is correctly computed by the wormhole.

For concreteness, consider the average of the following product of five-point functions

$$\overline{\langle \mathcal{O}_b \mathcal{O}_1 \mathcal{O}_a \mathcal{O}_4 \mathcal{O}_c \rangle \langle \mathcal{O}_b \mathcal{O}_1 \mathcal{O}_a \mathcal{O}_4 \mathcal{O}_c \rangle^*}.$$ (3.41)

This is associated with splitting the wormhole (3.35) along a five-punctured sphere in the bulk. The average (3.41), which corresponds to the two-boundary sphere five-point function wormhole, is given by the corresponding five-point function in Liouville CFT as in (3.24). In the Gaussian ensemble this however requires that we expand the two five-point functions in aligned channels when taking the ensemble average. Of course we are free to expand the five-point functions in different channels, in which case we need to invoke the non-Gaussian statistics. The combination of OPE channels that is associated to the particular Heegaard splitting is determined by the combination of sphere three-point boundaries that appear in each compression body of the Heegaard splitting. For example, if we compute the averaged product of sphere five-point functions by expanding in the following channel where there is

not a Gaussian contraction

$$\overline{\langle \mathcal{O}_b \mathcal{O}_1 \mathcal{O}_a \mathcal{O}_4 \mathcal{O}_c \rangle \langle \mathcal{O}_b \mathcal{O}_1 \mathcal{O}_a \mathcal{O}_4 \mathcal{O}_c \rangle^*}$$

$$= \sum_{\mathcal{O}_{2,3,5,6}} \overline{c_{12a} c_{2b3} c_{3c4} c_{4a5}^* c_{56c}^* c_{61b}^*} \left| \text{(diagram)} (m_1) \quad \text{(diagram)} (m_2) \right|^2 \tag{3.42}$$

$$= \left| \int_0^\infty dP_3 \, dP_6 \, \rho_0(P_3) \rho_0(P_6) C_0(P_1, P_b, P_6) C_0(P_6, P_a, P_3) C_0(P_3, P_c, P_4) \right.$$

$$\times \left. \text{(diagram)} (m_1) \quad \text{(diagram)} (m_2) \right|^2 \tag{3.43}$$

$$= |Z_{\text{Liouville}}(P_b, P_1, P_a, P_4, P_c | m_1, m_2)|^2, \tag{3.44}$$

then making use of the sixth moment (3.40) and the fact that the $6j$ symbols implement crossing transformations on the conformal blocks, we reproduce exactly the result from the Gaussian contraction, the sphere five-point function in Liouville theory. Here $m_i$ collectively denote the moduli of each five-point function.

We could have considered other Heegaard splittings, corresponding to averaged CFT observables that receive contributions from this combination of structure constants in a particular OPE channel. For example, the following averaged product of three four-point functions

$$\overline{\langle \mathcal{O}_6 \mathcal{O}_1 \mathcal{O}_2 \mathcal{O}_3 \rangle \langle \mathcal{O}_1 \mathcal{O}_2 \mathcal{O}_4 \mathcal{O}_5 \rangle \langle \mathcal{O}_6 \mathcal{O}_5 \mathcal{O}_4 \mathcal{O}_3 \rangle} \tag{3.45}$$

receives contributions from the sixth moment (3.40) in a specific OPE channel that precisely reproduce the result (3.10) for the averaged product in the Gaussian ensemble.

### 3.3.2 A more nontrivial six-boundary example

Here we consider another wormhole with six three-punctured sphere boundaries, but with the defects arranged slightly differently between the boundaries

$$M = \qquad\qquad\qquad\qquad . \tag{3.46}$$

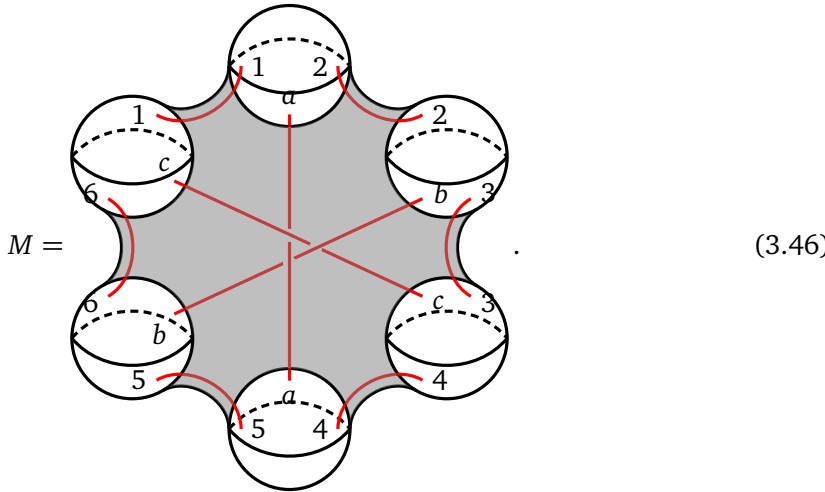

This contributes to a different sixth moment of the structure constants

$$|Z_{\text{Vir}}(M)|^2 \longleftrightarrow \overline{c_{1a2} c_{2b3} c_{3c4} c_{4a5} c_{5b6} c_{6c1}}. \tag{3.47}$$

We could of course compute the TQFT partition function on this wormhole by a straightforward Heegaard splitting, for example along three four-punctured spheres. In this case it turns out to be most convenient to replace the three-punctured sphere boundaries with trivalent junctions as in (2.26) and hence regard the wormhole as a network of Wilson lines embedded in $S^3$:

$$Z_{\text{Vir}}(M) = C_{12a}C_{23b}C_{34c}C_{45a}C_{56b}C_{61c}\, Z_{\text{Vir}}\left( \vcenter{\hbox{diagram}} \right) \tag{3.48}$$

$$\equiv C_{12a}C_{23b}C_{34c}C_{45a}C_{56b}C_{61c}\, Z_{\text{Vir}}(M'). \tag{3.49}$$

Here $M'$ is the network of Wilson lines depicted on the right-hand side of (3.48) embedded in $S^3$. Braiding the Wilson lines and applying a fusion transformation, the TQFT partition function may then be simplified as follows[18]

$$Z_{\text{Vir}}(M') = (\mathbb{B}_{P_4}^{P_a P_5}\mathbb{B}_{P_6}^{P_c P_1}\mathbb{B}_{P_4}^{P_3 P_c})^{-1}\, Z_{\text{Vir}}\left( \vcenter{\hbox{diagram}} \right) \tag{3.51}$$

$$= (\mathbb{B}_{P_4}^{P_a P_5}\mathbb{B}_{P_6}^{P_c P_1}\mathbb{B}_{P_4}^{P_3 P_c})^{-1} \int dP_d\, \mathbb{F}_{P_3 P_d}\begin{bmatrix} P_2 & P_c \\ P_b & P_4 \end{bmatrix} Z_{\text{Vir}}\left( \vcenter{\hbox{diagram}} \right). \tag{3.52}$$

We then recognize the following Wilson line identity (see [1, eq. (3.44)])

$$\vcenter{\hbox{diagram}} = \sqrt{\frac{C_{23t}}{C_{12s}C_{34s}C_{14t}}}\begin{Bmatrix} P_1 & P_2 & P_s \\ P_3 & P_4 & P_t \end{Bmatrix} \vcenter{\hbox{diagram}} \tag{3.53}$$

which allows us to recast the TQFT partition function as

$$Z_{\text{Vir}}(M') = (\mathbb{B}_{P_4}^{P_a P_5}\mathbb{B}_{P_6}^{P_c P_1}\mathbb{B}_{P_4}^{P_3 P_c})^{-1} \int dP_d\, \rho_0(P_d)\sqrt{\frac{C_{2cd}C_{a6d}}{C_{2b3}C_{3c4}C_{4a5}C_{5b6}}}$$

$$\times \begin{Bmatrix} P_2 & P_3 & P_b \\ P_4 & P_d & P_c \end{Bmatrix}\begin{Bmatrix} P_4 & P_5 & P_a \\ P_6 & P_d & P_b \end{Bmatrix} Z_{\text{Vir}}\left( \vcenter{\hbox{diagram}} \right). \tag{3.54}$$

---

[18]Here

$$\mathbb{B}_{P_i}^{P_j P_k} = e^{\pi i (P_i^2 - P_j^2 - P_k^2 - \frac{Q^2}{4})} \tag{3.50}$$

is the braiding phase.

Finally, we undo the crossings by braiding the Wilson lines and recognize the remaining configuration as the four-boundary wormhole studied in section 3.2 to arrive at

$$
Z_{\text{Vir}}(M) = \sqrt{C_{12a}C_{23b}C_{34c}C_{45a}C_{56b}C_{61c}}\, e^{\pi i(P_1^2+P_3^2+P_5^2-2P_2^2-2P_4^2-2P_6^2)}
$$
$$
\times \int dP_d\, \rho_0(P_d)\, e^{3\pi i P_d^2} \begin{Bmatrix} P_6 & P_1 & P_c \\ P_2 & P_d & P_a \end{Bmatrix} \begin{Bmatrix} P_2 & P_3 & P_b \\ P_4 & P_d & P_c \end{Bmatrix} \begin{Bmatrix} P_4 & P_5 & P_a \\ P_6 & P_d & P_b \end{Bmatrix}. \tag{3.55}
$$

Once the dust has settled, as in previous examples the wormhole partition function is given by factors of $\sqrt{C_0}$ for each sphere three-point boundary together with a suitable combination of Virasoro $6j$ symbols associated with the Wilson line crossings.

This wormhole partition function implies that the corresponding sixth moment for the CFT structure constants receives the following contribution

$$
\overline{c_{12a}c_{23b}c_{34c}c_{45a}c_{56b}c_{61c}} \supset \sqrt{\overline{c_{12a}^2}\,\overline{c_{23b}^2}\,\overline{c_{34c}^2}\,\overline{c_{45a}^2}\,\overline{c_{56b}^2}\,\overline{c_{61c}^2}}\,(-1)^{\ell_1+\ell_3+\ell_5}
$$
$$
\times \left| \int dP_d\, \rho_0(P_d) e^{3\pi i P_d^2} \begin{Bmatrix} P_6 & P_1 & P_c \\ P_2 & P_d & P_a \end{Bmatrix} \begin{Bmatrix} P_2 & P_3 & P_b \\ P_4 & P_d & P_c \end{Bmatrix} \begin{Bmatrix} P_4 & P_5 & P_a \\ P_6 & P_d & P_b \end{Bmatrix} \right|^2. \tag{3.56}
$$

As in previous examples, this sixth moment precisely affirms the internal consistency of the description in terms of an ensemble of CFT data. Indeed, if one expands for example the product of two sphere five-point functions or three sphere four-point functions in certain OPE channels where there is not a Gaussian contraction, this leads to a result consistent with the computation in the Gaussian ensemble. For concreteness, consider the following averaged product of three four-point functions, all expanded in the $u$-channel

$$
\overline{\langle \mathcal{O}_2 \mathcal{O}_c \mathcal{O}_a \mathcal{O}_6 \rangle \langle \mathcal{O}_6 \mathcal{O}_a \mathcal{O}_b \mathcal{O}_4 \rangle \langle \mathcal{O}_4 \mathcal{O}_b \mathcal{O}_c \mathcal{O}_2 \rangle}
$$
$$
= \sum_{\mathcal{O}_{1,3,5}} \overline{c_{1a2}c_{2b3}c_{3c4}c_{4a5}c_{5b6}c_{6c1}} \left| 2 \;\; \overset{c\;\;a}{\underset{1}{\times}} \;\; 6\; 6 \;\; \overset{a\;\;b}{\underset{5}{\times}} \;\; 4\; 4 \;\; \overset{b\;\;c}{\underset{3}{\times}} \;\; 2 \right|^2 \tag{3.57}
$$
$$
= \left| \int dP_1\, \rho_0(P_1)\, dP_3\, \rho_0(P_3)\, dP_5\, \rho_0(P_5) \sqrt{C_{1a2}C_{2b3}C_{3c4}C_{4a5}C_{5b6}C_{6c1}}\, e^{\pi i(P_1^2+P_3^2+P_5^2)} \right.
$$
$$
\times e^{-2\pi i(P_2^2+P_4^2+P_6^2)} \int dP_d\, \rho_0(P_d) e^{3\pi i P_d^2} \begin{Bmatrix} P_6 & P_1 & P_c \\ P_2 & P_d & P_a \end{Bmatrix} \begin{Bmatrix} P_2 & P_3 & P_b \\ P_4 & P_d & P_c \end{Bmatrix} \begin{Bmatrix} P_4 & P_5 & P_a \\ P_6 & P_d & P_b \end{Bmatrix}
$$
$$
\left. \times \; 2 \;\; \overset{c\;\;a}{\underset{1}{\times}} \;\; 6\; 6 \;\; \overset{a\;\;b}{\underset{5}{\times}} \;\; 4\; 4 \;\; \overset{b\;\;c}{\underset{3}{\times}} \;\; 2 \right|^2 \tag{3.58}
$$
$$
= \left| \int dP_d\, \rho_0(P_d) C_{2cd}C_{4bd}C_{6ad} \; 2 \;\; \overset{c\;\;a}{\underset{d}{\shortmid\shortmid}} \;\; 6\; 6 \;\; \overset{a\;\;b}{\underset{d}{\shortmid\shortmid}} \;\; 4\; 4 \;\; \overset{b\;\;c}{\underset{d}{\shortmid\shortmid}} \;\; 2 \right|^2. \tag{3.59}
$$

So we see that applying the statistics (3.56) precisely reproduces the result (3.10) anticipated from the Gaussian ensemble.

Notice that in this case the corresponding sixth moment receives contributions from configurations in which the Wilson lines have a different pattern of over- and under-crossings in the bulk, in addition to those with higher topology in the bulk. In principle, we could consider contributions from the manifolds formed by cutting $M$ along a six-punctured sphere in the bulk and gluing in another six-punctured sphere with any tangle formed by three strands in

the bulk. As a simple example, we could have considered the following six-boundary wormhole

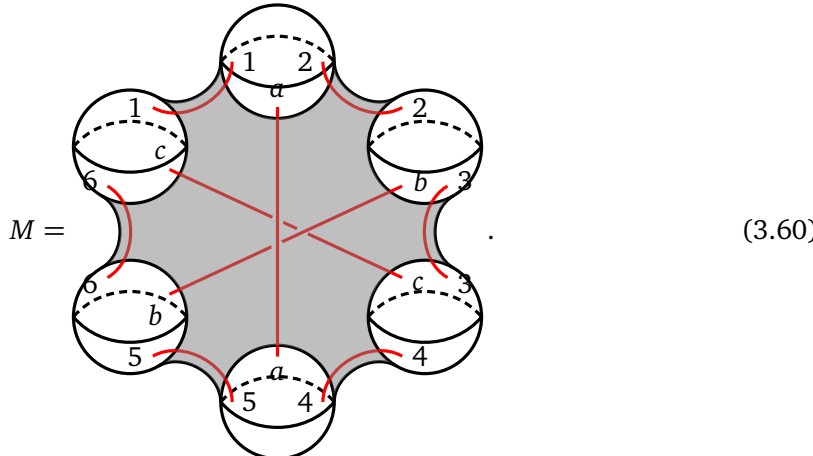

$$M = \qquad\qquad . \tag{3.60}$$

The TQFT partition function on this wormhole differs from (3.55) in a subtle way

$$Z_{\text{Vir}}(M) = \sqrt{C_{1a2}C_{2b3}C_{3c4}C_{4a5}C_{5b6}C_{6c1}}\, e^{\pi i(-P_1^2 + P_3^2 + P_5^2 - 2P_4^2)}$$

$$\times \int dP_d\, \rho_0(P_d)\, e^{\pi i P_d^2} \begin{Bmatrix} P_6 & P_1 & P_c \\ P_2 & P_d & P_a \end{Bmatrix} \begin{Bmatrix} P_2 & P_3 & P_b \\ P_4 & P_d & P_c \end{Bmatrix} \begin{Bmatrix} P_4 & P_5 & P_a \\ P_6 & P_d & P_b \end{Bmatrix}. \tag{3.61}$$

The only difference from (3.55) are the phases, particularly that which appears in the integral over the intermediate Liouville momentum $P_d$. Although both contribute to the corresponding sixth moment of the structure constants, between (3.55) and (3.61) is not a priori obvious which Wilson line configuration dominates in the semiclassical limit.

### 3.3.3 Diagrammatic rules for multi-boundary wormholes and CFT statistics

Although the intermediate details of the computations were nontrivial, there is an underlying simplicity to the previously discussed results for the Virasoro TQFT partition functions of wormholes with three-punctured sphere boundaries and trivial topology in the bulk, and hence for the leading contributions to the non-Gaussian statistics of CFT data in the boundary ensemble description of 3d gravity. In all cases, the wormhole partition function involves a factor of $\sqrt{C_0}$ for each three-punctured sphere boundary, together with a suitable combination of Virasoro $6j$ symbols. Here we describe diagrammatic rules that straightforwardly reproduce these results and that enable the computation of more nontrivial wormhole partition functions. These rules will turn out to be a slight generalization of the disk Feynman rules in JT gravity coupled to matter (see e.g. [21]).[19]

It is simplest to describe the situation in which the sphere boundaries are connected in a cyclic way, as in (3.30) and (3.46); the CFT statistics in other configurations may be obtained from the results in these cases by application of the swapping rule (3.29).

The idea is the following. Starting from a wormhole configuration with the boundaries connected in a cyclic way, replacing the punctured sphere boundaries with a trivalent vertex as follows

$$\tag{3.62}$$

---

[19]SC is grateful to Baur Mukhametzhanov for discussions on this. Baur also independently observed that higher moments of CFT data required for internal consistency of the ensemble description of 3d gravity were reproduced by generalizations of the disk Feynman diagrams in JT gravity coupled to matter [40].

produces a disk diagram with lines that may cross in the interior of the disk, such as that drawn in (3.48). It is important to keep track of the way that the lines over- and under-cross in the projection to a two-dimensional disk diagram. The TQFT partition function associated with this disk diagram is then computed according to the following simple Feynman rules:

- Each trivalent vertex contributes a factor of $\sqrt{C_0}$:

$$\text{(diagram)} = \sqrt{C_0(P_1, P_2, P_3)}. \tag{3.63}$$

- Each closed region in the interior of the disk is associated with a Liouville momentum $P$ that is integrated with the measure $\rho_0(P)\mathrm{d}P$.

- Each crossing of a pair of lines in the interior of the disk contributes a Virasoro $6j$ symbol

$$\text{(diagram)} = \begin{Bmatrix} P_1 & P_2 & P_s \\ P_3 & P_4 & P_t \end{Bmatrix} e^{\pi i (P_1^2 + P_3^2 - P_2^2 - P_4^2)}. \tag{3.64}$$

Here the labels 1, 2, 3 and 4 are associated to the four faces delineated by the Wilson lines $s$ and $t$.

The Virasoro $6j$ symbol plays the role of a quartic vertex in these diagrammatic rules, dressed with a phase that keeps track of the way that the Wilson lines over- and under-cross. This reproduces the partition function on the four-boundary wormhole (3.31) essentially by design.

As a simple example, consider the six-boundary wormhole studied in section 3.3.2. The two-dimensional projection of this configuration involves three crossings of Wilson lines and one closed region in the interior of the disk, so the TQFT partition function involves a single integral of three $6j$ symbols. Indeed, a straightforward application of these rules immediately reproduces the TQFT partition function (3.55).

The Virasoro $6j$ symbol obeys many identities that facilitate the consistency of this description. For instance, it is often the case that there is an ambiguity of how to arrange the Wilson line crossings in the interior of the disk. The TQFT partition function as computed from these rules should be independent of such choices. For example, we should have

$$\text{(diagram)} = \text{(diagram)}, \tag{3.65}$$

which is guaranteed by idempotency of the Virasoro $6j$ symbol

$$\int \mathrm{d}P_s \, \rho_0(P_s) \begin{Bmatrix} P_4 & P_1 & P_b \\ P_2 & P_s & P_a \end{Bmatrix} \begin{Bmatrix} P_2 & P_3 & P_a \\ P_4 & P_s & P_b \end{Bmatrix} = \frac{\delta(P_1 - P_3)}{\rho_0(P_1)}. \tag{3.66}$$

There is also a Yang-Baxter equation, which facilitates moving a line over a crossing as follows,

$$\text{(diagram)} = \text{(diagram)}. \tag{3.67}$$

In equations, this translates to

$$\int dP_d\, \rho_0(P_d) \begin{Bmatrix} P_1 & P_2 & P_a \\ P_3 & P_d & P_b \end{Bmatrix} \begin{Bmatrix} P_5 & P_6 & P_b \\ P_1 & P_d & P_c \end{Bmatrix} \begin{Bmatrix} P_3 & P_4 & P_c \\ P_5 & P_d & P_a \end{Bmatrix} e^{\pi i(P_d^2 + P_2^2 + P_4^2 + P_6^2)}$$

$$= \int dP_e\, \rho_0(P_e) \begin{Bmatrix} P_6 & P_1 & P_c \\ P_2 & P_e & P_a \end{Bmatrix} \begin{Bmatrix} P_2 & P_3 & P_b \\ P_4 & P_e & P_c \end{Bmatrix} \begin{Bmatrix} P_4 & P_5 & P_a \\ P_6 & P_e & P_b \end{Bmatrix} e^{\pi i(P_e^2 + P_1^2 + P_3^2 + P_5^2)}. \quad (3.68)$$

This identity follows from the consistency of braiding on the sphere. Indeed, the R-matrix also appears as the braiding matrix of conformal blocks as in eq. (3.33). The Yang-Baxter equation then corresponds to the fundamental relation in the braid group as follows:

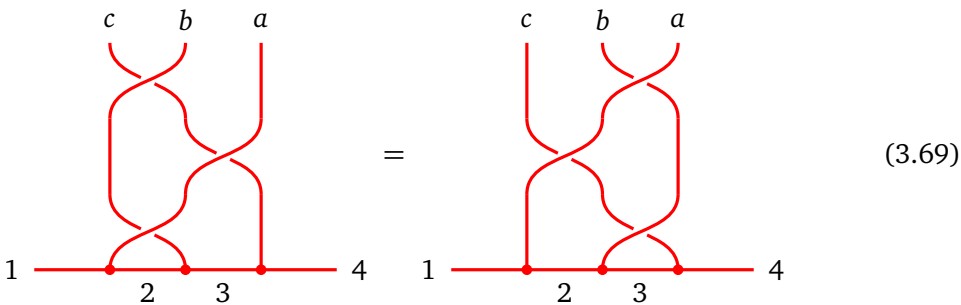

$$(3.69)$$

Using (3.33) to unbraid the left- and right-hand side and comparing the result leads to the Yang-Baxter equation (3.68).

These diagrammatic rules for wormhole partition functions are structurally identical to the disk Feynman rules for JT gravity coupled to matter, as described for example in [21]. The only differences are that here the trivalent vertex is given by $\sqrt{C_0}$, the quartic vertex is given by the Virasoro $6j$ symbol rather than the $SL(2, \mathbb{R})$ $6j$ symbol, and one must keep track of the over- and under-crossings of the Wilson lines in the bulk, leading to extra phases in the quartic vertex. Indeed, these rules precisely reduce to the JT gravity + matter disk Feynman rules in the semiclassical near-extremal limit of [5,41]. In this limit one takes

$$c = 1 + 6(b + b^{-1})^2, \qquad P_{\text{ext}} = b s_{\text{ext}}, \qquad P_{\text{int}} = \frac{i}{2}(b + b^{-1} - 2b h_{\text{int}}), \qquad b \to 0, \quad (3.70)$$

fixing $s_{\text{ext}}$ and $h_{\text{int}}$ in the semiclassical limit. Here $P_{\text{ext}}$ are the Liouville momenta of the Wilson lines forming the perimeter of the disk, $P_{\text{int}}$ are the Liouville momenta of those in the interior of the disk, and this limit corresponds to sending the external Wilson lines very near extremality while assigning the internal Wilson lines a fixed conformal weight $h_{\text{int}}$. With all external Wilson lines near extremality, the extra phase in the quartic vertex (3.64) cancels and we no longer need to keep track of the over- and under-crossing of the Wilson lines in the semiclassical limit.

It is likely that these diagrammatic rules may be derived directly from the tensor model for AdS$_3$ gravity recently introduced in [11], with tensor model diagrams corresponding to specific wormhole topologies. However we will not pursue this any further here.

## 3.4 Handle wormholes

In [10], the on-shell action of a class of wormholes contributing to certain single-boundary observables was constructed. These wormholes admitted an elegant interpretation in terms of the Coleman-Giddings-Strominger mechanism [42–44], whereby the existence of Euclidean wormholes induce random bulk couplings in the low-energy effective theory. Here we demonstrate that the gravity partition function on these single-boundary "handle wormholes" is straightforward to compute using Virasoro TQFT.

For concreteness, consider the sphere four-point function of pairwise identical operators $\langle \mathcal{O}_1 \mathcal{O}_2 \mathcal{O}_2 \mathcal{O}_1 \rangle$. Suppose there is a third species of defect, dual to the operator $\mathcal{O}_3$. Naively, the trivalent coupling $\lambda_{123}$ in the bulk low-energy effective field theory vanishes since in the Gaussian ensemble the averaged structure constant vanishes $\overline{c_{123}} = 0$. However, there is a two-boundary wormhole that computes the variance $\overline{c_{123}^2} \neq 0$, so the conclusion that the defects are entirely non-interacting in the bulk cannot quite be correct. In particular, we expect a topology that corresponds to the exchange of $\mathcal{O}_3$ in the $\mathcal{O}_1 \times \mathcal{O}_2$ OPE and hence contributes to the bulk-dual of the four-point function $\langle \mathcal{O}_1 \mathcal{O}_2 \mathcal{O}_2 \mathcal{O}_1 \rangle$.

Consider the following topology discussed in [10]

$$M = \qquad\qquad\qquad\qquad\qquad\qquad\qquad\qquad (3.71)$$

It is constructed by starting with a compression body whose outer boundary is a four-punctured sphere and two three-punctured sphere inner boundaries, and then identifying the two inner boundaries as shown in (3.71). The Wilson lines corresponding to $\mathcal{O}_1$ and $\mathcal{O}_2$ traverse the resulting wormhole and that corresponding to $\mathcal{O}_3$ forms a closed loop in it. The TQFT partition function on the compression body (without the identification among the inner boundaries) is simply proportional to the corresponding sphere four-point conformal block

$$Z_{\text{Vir}} \left( \vphantom{\Bigg|} \right) = C_0(P_1, P_2, P_3)^2 \qquad\qquad\qquad\qquad (3.72)$$

To implement the identification of the inner boundaries, we first view the partition function on the compression body as a state in the tensor product Hilbert space associated with the inner and outer boundaries $\mathcal{H}_{0,4} \otimes \mathcal{H}_{0,3} \otimes \mathcal{H}_{0,3}$. Taking the inner product between the states in the three-punctured sphere Hilbert spaces implements the identification between the inner boundaries and leaves us with the following state in $\mathcal{H}_{0,4}$:

$$Z_{\text{Vir}}(M) = C_0(P_1, P_2, P_3) \qquad\qquad\qquad\qquad (3.73)$$

Squaring the TQFT partition function leads to the expected contribution to the gravity path integral corresponding to the exchange of $\mathcal{O}_3$ in the $\mathcal{O}_1 \times \mathcal{O}_2$ OPE, with squared OPE coefficient given by the corresponding variance in the Gaussian ensemble

$$Z_{\text{grav}}(M) = |C_0(P_1, P_2, P_3)|^2 \left| \vphantom{\Bigg|} \right|^2 . \qquad\qquad\qquad (3.74)$$

This is precisely the result that was computed semiclassically in [10].

## 3.5 Twisted $I$-bundles

Let us discuss another interesting example which has appeared before in the literature on AdS$_3$ gravity known as a twisted $I$-bundle. It was studied in [45] as a simple example of a non-handlebody saddle-point contribution to the 3d gravity path integral with a single higher-genus boundary. The name stems from the fact that these three-manifolds are constructed as a non-trivial $I$-bundle over a Riemann surface, where $I$ is an interval. Consider a hyperbolic Riemann surface $\Sigma$ together with an orientation-reversing (i.e. anti-holomorphic) fixed-point free involution $\Phi : \Sigma \to \Sigma$. We can then consider a quotient of the Euclidean wormhole $\Sigma \times [0,1]$ as follows:

$$M_\Phi = (\Sigma \times [0,1])/\{(z,x) \sim (\Phi(z), 1-x)\}. \tag{3.75}$$

This identification is again orientation-preserving and thus we get an orientable hyperbolic manifold with a single boundary $\Sigma$, where the hyperbolic structure is inherited from the Euclidean wormhole.

$\Phi$ induces an involution on the boundary Teichmüller space which we also call $\Phi$ and hence the boundary moduli are constrained to lie on the fixed point set $\mathcal{T}^\Phi$. By the uniformization theorems of three-dimensional hyperbolic manifolds that we reviewed in the Appendix of [1], we are however guaranteed that the manifold with the same topology can also be defined away from the real locus in Teichmüller space. The construction then proceeds by taking a quotient of a quasi-Fuchsian wormhole, where the moduli of the left boundary are the image under $\Phi$ of the moduli of the right boundary.

From the TQFT point of view, it is very simple to determine the Virasoro TQFT partition function on these manifolds. Indeed, we could squash the manifold to the surface $\Sigma \times \{\frac{1}{2}\}$ and the quotient by $\Phi$ simply produces $\widetilde{\Sigma} \times \{\frac{1}{2}\}$. Here,

$$\widetilde{\Sigma} = \Sigma/\{z \sim \Phi(z)\}, \tag{3.76}$$

is the non-orientable surface obtained from quotienting $\Sigma$. Given that the Virasoro TQFT partition function on the Euclidean wormhole is simply the Liouville partition function, we see that $\Phi$ acts precisely by an orientifold projection. In other words, the partition function on the twisted $I$-bundle is simply the Liouville partition function on the non-orientable surface $\widetilde{\Sigma}$.

To see that this makes sense, recall that the conformal block expansion on a non-orientable surface involves a single conformal block on the doubled surface $\Sigma$ which hence defines a state in the boundary Hilbert space of the twisted $I$-bundle. Let us make this more concrete by recalling the precise construction of Liouville theory on a non-orientable surface. We can construct a non-orientable surface by including a number of cross-caps on an orientable surface.[20] E.g. on a torus with one puncture and a cross-cap, we have



$$\tag{3.77}$$

The orientifold acting reflects the right hand side of the picture to the left side and simultaneously rotates by 180 degrees around the dashed horizontal line. This map has no fixed point and the quotient indeed leads to the crosscap state. On the level of the conformal blocks, this means that the conformal block of the Liouville partition function on this surface takes the form

---

[20]Since two crosscaps are equivalent to a handle in the presence of another crosscap, one can restrict to one or two crosscaps.

$$Z_L\left(\ \vcenter{\hbox{}}\ \right) = \int_0^\infty \mathrm{d}P_1\,\mathrm{d}P_2\,\mathrm{d}P_3\ \rho_0(P_1)\rho_0(P_2)\Gamma(P_3)C_0(P_0,P_1,P_2)$$

$$\times\ C_0(P_1,P_2,P_3)\ \vcenter{\hbox{}} \tag{3.78}$$

where the picture represents the ordinary conformal block. The only new ingredient is the normalization of the crosscap state given by $\Gamma(P_3)$. It is fully determined by requiring consistency with the bootstrap. It is in general given by [46]

$$\Gamma(P) = \frac{\mathbb{P}_{\mathbb{1},P}}{\sqrt{\mathbb{S}_{\mathbb{1},P}}} \times \sqrt{\rho_0(P)} = \mathbb{P}_{\mathbb{1},P}\,. \tag{3.79}$$

Here, the first factor is the general result when the two-point function of the theory is canonically normalized. We then multiply by $\sqrt{\rho_0(P)}$ to account for our normalization of the two-point function. The $\mathbb{P}$-matrix describes the modular transformation of the Möbius strip characters:

$$\mathbb{P} = \mathbb{T}^{\frac{1}{2}}\mathbb{S}\mathbb{T}^2\mathbb{S}\mathbb{T}^{\frac{1}{2}}\,. \tag{3.80}$$

It is simple to work this out explicitly:

$$\mathbb{P}_{P_1,P_2} = 8\int \mathrm{d}P\ e^{\pi i(P_1^2+P_2^2+4P^2-\frac{1}{4})}\cos(4\pi P_1 P)\cos(4\pi P_2 P) = 2\cos(2\pi P_1 P_2)\,. \tag{3.81}$$

Thus we have

$$\Gamma(P) = \mathbb{P}_{\mathbb{1},P} = \mathbb{P}_{P_1=\frac{i(b^2+1)}{2b},P} + \mathbb{P}_{P_1=\frac{i(b^2-1)}{2b},P} = 4\cosh(\pi b P)\cosh(\pi b^{-1}P)\,. \tag{3.82}$$

The $+$ sign comes from a careful treatment of the factor $\mathbb{T}^{\frac{1}{2}}$ in the definition of the $\mathbb{P}$-matrix; more physically, it comes because the orientifold projection acts by a factor $(-1)^N$ on a level $N$ descendant. This is the same result as obtained in [47,48] after translating to our conventions. This fully specifies the Liouville partition function on any non-orientable surface and hence directly gives the value of $Z_{\mathrm{Vir}}$ on any twisted $I$-bundle.

Finally, the gravity partition function is given by applying eq. (1.1),

$$Z_{\mathrm{grav}}(M_\Phi) = \sum_{\gamma\in\mathrm{Map}(\Sigma)/\mathrm{Map}(\widetilde{\Sigma})} |Z_L(\widetilde{\Sigma}^\gamma)|^2\,, \tag{3.83}$$

where we used that the bulk mapping class group is the mapping class group of the non-orientable surface $\widetilde{\Sigma}$ under which the Liouville partition function is invariant by crossing symmetry.

# 4 The figure eight knot complement

In this section, we look at one particular hyperbolic 3-manifold in detail and illustrate some features of the theory through this example. The manifold in question is the figure eight knot complement, i.e. $\mathrm{S}^3$ with a Wilson line inside forming a figure eight knot. This manifold is known to admit a hyperbolic metric. The figure eight knot is the hyperbolic knot with the smallest possible volume and the only knot with the crossing number 4, as demonstrated

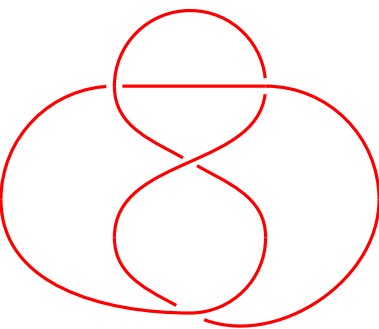

Figure 9: A visualization of the figure eight knot.

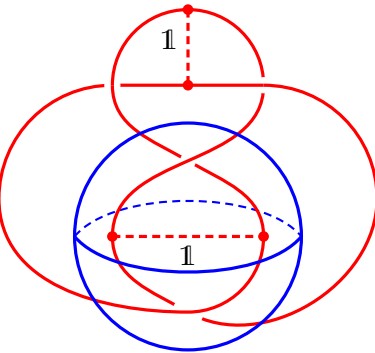

Figure 10: Heegaard splitting of the figure eight knot complement.

in Figure 9. Thus it is usually denoted as $4_1$. There are two approaches to calculate the Virasoro TQFT partition function of the figure eight knot complement. One way to compute the partition function is via the Heegaard splitting procedure. The other way is to consider the surface bundle construction of the figure eight knot, and to use the mapping torus technique introduced in [1]. These two approaches will lead to different integral expressions as the final results. We check that these two expressions agree and both have the same semiclassical expansions as expected.

## 4.1 Direct computation

Let us first compute the partition function by successively undoing the over- and under-crossings in a particular projection of the knot.

We start by computing the partition function via surgery. We embed the above knot configuration into a three-sphere to create the figure eight knot complement. In the TQFT setup, we consider the knot as a tangled Wilson loop with associated conformal weight $\Delta_0 = \frac{Q^2}{4}$, i.e. the cusp, although we will keep the label of the Wilson loop generic for most of the discussion. If we slice the above figure 9 into halves along the equatorial $S^2$, we obtain two manifolds $M_1, M_2$ with boundaries as four-punctured sphere. The path integral over each half prepares a state in the Hilbert space $\mathcal{H}_{\Sigma_{0,4}}$, and the partition function is the inner product between these two sphere 4-point conformal blocks. Here the Wilson lines inside each component have nontrivial braidings. Before evaluating the inner product, we want to untangle the Wilson lines. For this purpose, we need to apply the crossing and braiding operations on the boundary surface $\Sigma_{0,4}$.

To make the crossing and braiding explicit, we firstly specify the intermediate channels in the figure eight knot. In the diagram, we have identity operators propagate in the intermediate channels corresponding to the contractible cycles in the bulk. We can use the fusion kernel $\mathbb{F}$

to transform the diagram 10 into the other channel

$$
Z_{\text{Vir}}\left(\text{}\right) = \int dP_s\, dP_t\, \mathbb{F}_{1,P_s}\begin{bmatrix} P_0 & P_0 \end{bmatrix} \mathbb{F}_{1,P_t}\begin{bmatrix} P_0 & P_0 \end{bmatrix}
$$

$$
\times Z_{\text{Vir}}\left(\quad\right), \quad (4.1)
$$

where $P_0$ labels the conformal weight of the Wilson loop, i.e. $P_0 = 0$ for $\Delta = Q^2/4$. After transforming the figure eight knot diagram into the other channel, we can untangle the knot at each trivalent node via the braiding move $\mathbb{B}$ as follows

$$
Z_{\text{Vir}}(4_1) = \int dP_s\, dP_t\, \mathbb{F}_{1,P_s}\begin{bmatrix} P_0 & P_0 \end{bmatrix} \mathbb{F}_{1,P_t}\begin{bmatrix} P_0 & P_0 \end{bmatrix} (\mathbb{B}_{P_s}^{P_0,P_0})^2
$$

$$
\times (\mathbb{B}_{P_t}^{P_0,P_0})^{-2} Z_{\text{Vir}}\left(\quad\right). \quad (4.2)
$$

The fusion kernel $\mathbb{F}$ corresponding to the exchange of the identity operator can be written in terms of $\rho_0$ and $C_0$ as follows

$$
\mathbb{F}_{1,P}\begin{bmatrix} P_0 & P_0 \\ P_0 & P_0 \end{bmatrix} = \rho_0(P)C_0(P,P_0,P_0). \quad (4.3)
$$

Meanwhile, we recognize the remaining contraction as the four-boundary wormhole discussed in Section 3.2 for which we can use the result (3.20), normalized by inverse structure constants to account for the normalization of the junctures.

In the end, we obtain an integral expression of the figure eight knot partition function

$$
Z_{\text{Vir}}(4_1) = \int dP_s\, dP_t\, \rho_0(P_s)\rho_0(P_t)(\mathbb{B}_{P_s}^{P_0,P_0})^2(\mathbb{B}_{P_t}^{P_0,P_0})^{-2} \begin{Bmatrix} P_0 & P_0 & P_s \\ P_0 & P_0 & P_t \end{Bmatrix}. \quad (4.4)
$$

There are two momentum integrals in the above formula (4.4), and we can reduce the number of integrals by one by using the relation (2.33) between the fusion kernel $\mathbb{F}$ and the modular S-matrix $\mathbb{S}$. We hence get

$$
Z_{\text{Vir}}(4_1) = \int_0^\infty dP\, \frac{\rho_0(P)}{\rho_0(P_0)} e^{\frac{\pi i Q^2}{4} - 3\pi i P^2} \mathbb{S}_{P_0,P_0}[P] \quad (4.5)
$$

$$
= \int_0^\infty dP\, \frac{\rho_0(P) e^{\frac{3\pi i Q^2}{8} - \frac{5\pi i P^2}{2}}}{S_b(\frac{Q}{2} + iP)} \int_{-\infty}^\infty dx\, e^{-4\pi i x P_0} S_b(\tfrac{Q}{4} + \tfrac{iP}{2} \pm iP_0 \pm ix), \quad (4.6)
$$

where we inserted the explicit expression for the modular crossing kernel in the second line [18].

Of course, this expression is dependent on the framing that we implicitly chose in this computation. For the figure eight knot complement, a nice way to fix the framing anomaly is by requiring that the partition function should be real. Indeed, complex conjugation corresponds to orientation reversal, but since the figure eight knot is invariant under orientation reversal (this property is called amphichirality), we can choose the partition function to be real.

One can easily check, for example numerically, that this is the case if we multiply the above expression with $e^{4\pi i P_0^2} = e^{4\pi i(\Delta_0 - \frac{c}{24})}$, which is part of the ambiguity from framing. We hence have

$$Z_{\mathrm{Vir}}(4_1) = \int_0^\infty dP \, \frac{\rho_0(P) e^{\frac{3\pi i Q^2}{8} + 4\pi i P_0^2 - \frac{5\pi i P^2}{2}}}{S_b(\frac{Q}{2} + iP)} \int_{-\infty}^\infty dx \, e^{-4\pi i x P_0} S_b(\tfrac{Q}{4} + \tfrac{iP}{2} \pm iP_0 \pm ix), \quad (4.7)$$

which is the formula we will use from now on.

**Choice of contour.** There is one additional subtlety with this formula. As it stands, the integral over $P$ is actually not convergent. Indeed, using the asymptotics of the double sine function, see e.g. [26, eq. (B.53)] and using that the integral over $x$ is dominated for small $x$, we see that

$$\int_{-\infty}^\infty dx \, e^{-4\pi i x P_0} S_b(\tfrac{Q}{4} + \tfrac{iP}{2} \pm iP_0 \pm ix) \sim \frac{1}{\sqrt{2}} e^{\frac{\pi i P^2}{2} - \frac{\pi PQ}{2} + \frac{\pi i Q^2}{24} - \frac{\pi i}{12}}. \quad (4.8)$$

Combining this with the asymptotics of the rest of the integrand, we see that the integrand behaves for large $\mathrm{Re}(P)$ as

$$\mathrm{integrand}(P) \sim e^{\frac{3\pi QP}{2} - \frac{5\pi i P^2}{2}} \times \mathcal{O}(\text{order 1 in } P). \quad (4.9)$$

Thus the integral in (4.7) doesn't converge for $P$ on the real axis. However, we see that we could have improved convergence by taking $P$ to run along a contour starting at $P = 0$ and asymptoting for large $P$ the line $\mathbb{R} - ia$, where the shift $a$ has to be at least $a > \frac{3Q}{10}$ to ensure convergence. Shifting the contour in this way doesn't cross any poles and is hence a generally harmless operation. Thus it is understood that the integral over $P$ in (4.7) actually follows this modified contour.

## 4.2 Comparison to Teichmüller TQFT

The figure eight knot partition function can also be obtained in Teichmüller TQFT developed in [49–52]. Translating to our conventions, the expression for the Teichmüller TQFT partition function is[21]

$$Z_{\mathrm{Teich}}(4_1) = \sqrt{2} \int_{\mathbb{R} - i0^+} dx \, S_b(ix \pm 2iP_0). \quad (4.10)$$

Here the integral runs slightly below the real axis to avoid the poles at $x = \pm 2P_0$. This formula can be obtained by realizing the figure eight knot complement as a gluing of two tetrahedra. Each tetrahedron gives rise to one double sine function and the gluing to the integral (modulo some constraints).

---

[21]Teichmüller TQFT depends on a parameter $\hbar$, which, following the conventions of [50], we identify as $\hbar = -i\pi b^2$. This expression does not literally match the one given in [49–52]. We are unsure whether this is a typo in the previous literature. In any case, the semiclassical expansion that we discuss below *does* match previous expressions, which gives us a lot of confidence in the correctness of (4.10). We thank Boris Post and Davide Saccardo for discussions about this.

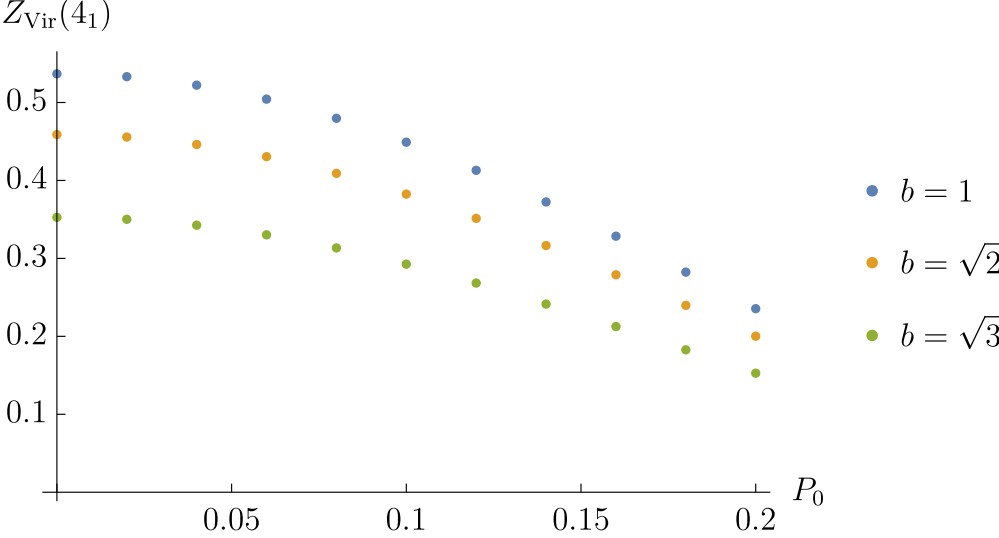

Figure 11: Numerical comparison of the Virasoro TQFT and the Teichmüller TQFT partition function of the figure eight knot complement. The plotted data points are for the Teichmüller expression (4.10), but are indistinguishable from the Virasoro data points.

As we already conjectured in our previous paper [1], we expect that Virasoro TQFT is equivalent to Teichmüller TQFT and thus the two expressions should match,

$$Z_{\text{Vir}}(4_1) \stackrel{!}{=} Z_{\text{Teich}}(4_1). \tag{4.11}$$

This equality turns out to be quite hard to prove analytically. However, we checked numerically for various values of $b$ and $P_0$ that the two expressions agree.

The numerical evaluation is in principle straightforward. We restricted our attention to rational values of $b^2$, since in this case, there is a simple way to express the double sine function through the Barnes G-function for which we can use efficient implementations, for example in `Mathematica`,

$$S_b(z) = (2\pi)^{\sqrt{mn}z - \frac{m+n}{2}} \prod_{k=0}^{m-1} \prod_{\ell=0}^{n-1} \frac{G\left(\frac{k+1}{m} + \frac{\ell+1}{n} - \frac{z}{\sqrt{mn}}\right)}{G\left(\frac{k}{m} + \frac{\ell}{n} + \frac{z}{\sqrt{mn}}\right)}. \tag{4.12}$$

It is then simple to compute the required integrals in (4.7) over a converging contour and compare with the simpler expression (4.10). We computed the partition functions for $b = 1$, $b = \sqrt{2}$ and $b = \sqrt{3}$ for $P_0 = 0, 0.02, \ldots, 0.2$. To the precision we have computed, all values agree to seven decimal places, thus showing the equality (4.11) beyond reasonable doubt. The data points are plotted in Figure 11.

From this discussion, it may seem that the Teichmüller TQFT always produces simpler expressions than Virasoro TQFT, but this is not the case. The expressions in Teichmüller TQFT become more complicated when the 3-manifold in question requires more tetrahedra to form a triangulation, while this is not necessarily so in Virasoro TQFT. It is in general quite hard to recognize when two integral representations of the partition function agree since there are an enormous number of non-trivial integral identities relating them.

## 4.3 Computation via the Seifert surface

Let us explain a completely different way to compute the partition function that will lead to an inequivalent integral for the partition function.

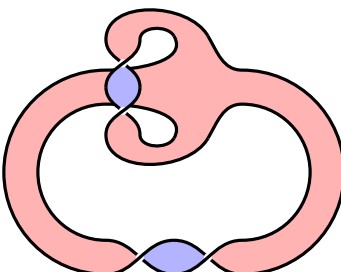

Figure 12: The Seifert surface of the figure eight knot. One can easily verify that the boundary of the Seifert surface traces out a figure eight knot.

The figure eight knot admits a genus 1 Seifert surface. This means that we can realize the knot as the boundary of a one-holed torus embedded in $S^3$, so that the boundary of the one-holed torus coincides with knot. This is depicted in Figure 12. However, even more is true. One can slightly deform the Seifert surface and obtain a foliation of the knot complement in terms of one-holed tori. The figure eight knot complement is in fact a surface bundle over a circle, i.e. it is of the form

$$[0,1] \times \Sigma_{1,1}/\sim, \tag{4.13}$$

where $\Sigma_{1,1}$ is the one-holed torus and we identify

$$(0,z) \sim (1,\phi(z)), \tag{4.14}$$

with $\phi = ST^3$ being the corresponding mapping class group element in $\mathrm{SL}(2,\mathbb{Z})$ generated by $S$ and $T$.

This might let one suspect that we can compute the partition function of the figure eight knot complement as

$$Z_{\mathrm{Vir}}(4_1) \overset{?}{=} \mathrm{tr}_{\mathcal{H}_{1,1}}(\mathbb{S}[P_0]\,\mathbb{T}^3), \tag{4.15}$$

but this is not quite correct yet. Indeed, taking the trace over the Hilbert space $\mathcal{H}_{1,1}$ of conformal blocks on the once-punctured torus would lead to the partition function of the three-dimensional manifold where the Wilson line runs along the thermal circle $S^1$. This is not what we want, since the Wilson line bounds the Seifert surface, which forms the meridian of the boundary torus of the manifold. This means that the correct expression is obtained by applying the S-modular transformation in the external parameter $P_0$. So we conclude that we should have

$$Z_{\mathrm{Vir}}(4_1) = \int_0^\infty \mathrm{d}P_0'\,\mathbb{S}_{P_0,P_0'}[\mathbb{1}]\,\mathrm{tr}_{\mathcal{H}_{1,1}}(\mathbb{S}[P_0']\,\mathbb{T}^3). \tag{4.16}$$

We can easily plug in the explicit expressions for the modular crossing kernel and get an alternative expression for the partition function of the figure eight knot complement. This expression is even more unwieldy then the previous ones, since it involves three integrals, one from the definition of $\mathbb{S}$, one from the trace, and one from the integral over $P_0'$. This pushes our numerical capabilities a bit too far. Instead, we will check below that the first two terms in the semiclassical expansion agree with the semiclassical expansion of the previous expression.[22]

We note that this expression makes reality of the partition function manifest, while it was obscured in the expression (4.7) that we discussed above. Indeed, one of the Moore-Seiberg relations states that $\mathbb{T}\mathbb{S}\mathbb{T}\mathbb{S}\mathbb{T} = \mathbb{S}$ as operators (see [1, eq. (A.5b)]) and thus

$$\mathrm{tr}(\mathbb{S}\mathbb{T}^3)^* = \mathrm{tr}(\mathbb{T}^{-1}\mathbb{S}^{-1}\mathbb{T}^{-2}) = \mathrm{tr}(\mathbb{S}\mathbb{T}\mathbb{S}^{-1}\mathbb{T}^{-1}) = \mathrm{tr}(\mathbb{S}\mathbb{T}^2\mathbb{S}\mathbb{T}\mathbb{S}^{-1}) = \mathrm{tr}(\mathbb{S}\mathbb{T}^3), \tag{4.17}$$

and so after Fourier transformation we still get a real function.

---

[22]We also checked that the corresponding expressions for the figure eight knot partition function in $\mathrm{SU}(2)_k$ Chern-Simons theory agree where all the integrals are just finite sums.

### 4.4 Semiclassical expansion

We now write down the semiclassical expansion of the Virasoro TQFT partition function in the form (4.10) and check the volume conjecture explicitly. This was already done before in the context of Teichmüller TQFT [49, 50] and hence we shall be rather brief.

The key identity is the semiclassical expansion of the double sine function,

$$\log S_b(x_0 + x) = \sum_{n=0}^{\infty} \frac{(2\pi i b^2)^{n-1}}{2n!} \Big( \mathrm{Li}_{2-n}\big(e^{-2\pi i b x_0}\big) - (-1)^n \mathrm{Li}_{2-n}\big(e^{2\pi i b x_0}\big) \Big) B_n\big(1 - \tfrac{x}{b}\big). \quad (4.18)$$

In this identity we think of $x_0$ as being of order $\mathcal{O}(\frac{1}{b})$, while $x$ is of order $\mathcal{O}(1)$. This identity is standard for the quantum dilogarithm to which the double sine function is closely related, see e.g. [52, Proposition 6]. For completeness, we have included a short derivation in appendix A.

We now apply this expansion as follows. In the semiclassical limit, the argument of the double sine function in eq. (4.10) becomes large and we write $P_0 = \frac{\eta_0}{b}$. We can then evaluate the integral via saddle point approximation. We write $x$ to leading order as $\frac{x_0}{b}$. Then the saddle-point equation is

$$0 = -\frac{1}{2\pi} \partial_{x_0} \sum_{\pm} \Big( \mathrm{Li}_2(e^{2\pi(x_0 \pm 2\eta_0)}) - \mathrm{Li}_2(e^{-2\pi(x_0 \pm 2\eta_0)}) \Big) \quad (4.19)$$

$$= \log \prod_{\pm, \pm} \big( 1 - e^{2\pi(\pm x_0 \pm 2\eta_0)} \big). \quad (4.20)$$

The solution to this saddlepoint equation takes the form

$$x_0 = \frac{1}{2\pi} \log\Big( \cosh(4\pi\eta_0) \pm_1 \tfrac{1}{2} \pm_2 \sqrt{\big( \cosh(4\pi\eta_0) \pm_1 \tfrac{1}{2} \big)^2 - 1} \Big) + ni, \quad n \in \mathbb{Z}. \quad (4.21)$$

The steepest descent contour runs through the saddle point at

$$x_0 = \frac{1}{2\pi} \log\Big( \cosh(4\pi\eta_0) - \tfrac{1}{2} - \sqrt{\big( \cosh(4\pi\eta_0) - \tfrac{1}{2} \big)^2 - 1} \Big), \quad (4.22)$$

and hence only that one is relevant for our analysis. For this to be valid, we should assume that

$$|\eta_0| < -\frac{1}{2\pi} \log\Big( \frac{\sqrt{5}-1}{2} \Big), \quad (4.23)$$

since otherwise $x_0$ becomes real and the saddle-point evaluation is different. We obtain the semiclassical expansion

$$Z_{\mathrm{Vir}}(4_1) = \frac{\sqrt{2}\, e^{-\frac{1}{2\pi b^2} \mathrm{vol}(4_1)}}{\Delta^{\frac{1}{4}}} \exp\Big( \sum_{n=1}^{\infty} S_n b^{2n} \Big), \quad (4.24)$$

where

$$\Delta = -X^2 + 2X + 1 + 2X^{-1} - X^{-2}, \qquad X = e^{4\pi\eta_0}, \quad (4.25)$$

and

$$\mathrm{vol}(4_1, \eta_0) = -\frac{i}{2}\Big( \mathrm{Li}_2\big(\tfrac{X-1+X^{-1}+i\sqrt{\Delta}}{2X}\big) + \mathrm{Li}_2\big(\tfrac{X-1+X^{-1}+i\sqrt{\Delta}}{2}X\big)$$
$$- \mathrm{Li}_2\big(\tfrac{X-1+X^{-1}-i\sqrt{\Delta}}{2X}\big) - \mathrm{Li}_2\big(\tfrac{X-1+X^{-1}-i\sqrt{\Delta}}{2}X\big) \Big). \quad (4.26)$$

The first few orders for the higher loop corrections are given by

$$S_1 = -\frac{\pi}{12\Delta^{\frac{3}{2}}}(X^3 - X^2 - 2X^2 + 15 - 2X^{-1} - X^{-2} + X^{-3}), \tag{4.27a}$$

$$S_2 = \frac{2\pi^2}{\Delta^3}(X^3 - X^2 - 2X + 5 - 2X^{-1} - X^{-2} + X^{-3}), \tag{4.27b}$$

$$S_3 = \frac{\pi^3}{90\Delta^{\frac{9}{2}}}(X^8 - 4X^7 - 128X^6 + 36X^5 + 1074X^4 - 5630X^3 + 5782X^2$$
$$+ 7484X^1 - 18311 + 7484X^{-1} + 5782X^{-2} - 5630X^{-3} + 1074X^{-4} + 36X^{-5}$$
$$- 128X^{-6} - 4X^{-7} + X^{-8}). \tag{4.27c}$$

Not surprisingly, this reproduces the semiclassical expansion given in [49–51]. Also noticed there, the one-loop determinant equals the Reidemeister torsion of the figure eight knot, which can also be derived from computing the functional analytic one-loop determinants appearing in 3d gravity. Thus this shows the validity of the volume conjecture (2.1) for the figure eight knot.

**Expression from Seifert surface.** We now reproduce the semiclassical expansion from the expression that we got from the computation via the Seifert surface as described in section 4.3. This gives strong evidence that the expression (4.16) is in fact equal to the simpler expression given by eq. (4.10).

By using the explicit formula of the Virasoro crossing kernel shown in [1], we rewrite the integral formula (4.16) in terms of double-sine functions

$$Z_{\text{Vir}}(4_1) = 2\sqrt{2}\int_0^\infty \mathrm{d}P_0' \cos 4\pi P_0 P_0' \int_0^\infty \mathrm{d}P\, \rho_0(P)$$
$$\times \int_{-\infty}^\infty \mathrm{d}\xi\, \frac{e^{\frac{\pi i \Delta_0'}{2} - 6\pi i (P^2 - \frac{1}{24}) - 4\pi i \xi P}}{S_b(\frac{Q}{2} + iP_0')} S_b(\tfrac{Q}{4} + \tfrac{iP_0'}{2} \pm iP \pm i\xi). \tag{4.28}$$

For simplicity, in the following computation, we consider $P_0 = 0$ which sets the conformal weight of the knot to be $\Delta = \frac{Q^2}{4}$. As we will see later, the saddle-point equation in the semi-classical approximation will be simplified in this case. In general, we can also compute the partition function for the knot with a generic conformal weight, while the complexity of solving the saddle-point equations increases. Once we consider the semiclassical limit of this expression, we similarly rescale $P_0' = \frac{\eta_0}{b}$, $P = \frac{x}{b}$ and $\xi = \frac{\eta}{b}$. Then we apply the expansion formula of the double sine function to write the integrand into a expansion in $1/b^2$.

$$Z_{\text{Vir}}(4_1) = \int \frac{\mathrm{d}\eta_0\, \mathrm{d}x\, \mathrm{d}\eta}{b^3} e^{\sum_{n=0}^\infty S^{(n)} b^{2(n-1)}}. \tag{4.29}$$

In $b \to 0$ limit, we can approximate this integral by saddle-point. The leading order contribution is proportional to $1/b^2$ with the coefficient

$$S^{(0)} = \frac{\pi i}{8} + 2\pi x + \frac{\pi i \eta_0^2}{2} - 6\pi i x^2 - 4\pi i \eta x - \frac{i}{4\pi}\left(\text{Li}_2(e^{-2\pi\eta_0 + i\pi}) - \text{Li}_2(e^{2\pi\eta_0 - i\pi})\right)$$
$$+ \frac{i}{4\pi}\sum_{\pm,\pm}\left(\text{Li}_2(e^{-2\pi(\frac{\eta_0}{2} \pm x \pm \eta) + \frac{\pi i}{2}}) - \text{Li}_2(e^{2\pi(\frac{\eta_0}{2} \pm x \pm \eta) - \frac{\pi i}{2}})\right). \tag{4.30}$$

This leads to three saddle-point equations.

Since $\mathbb{S}[P'_0]$ only depends on the conformal weight $\Delta'_0 = P'^2_0 + \frac{Q^2}{4}$, the function $\mathrm{tr}(\mathbb{S}[P'_0]\mathbb{T}^3)$ is even in $P'_0$. This observation implies that $\eta_0 = 0$ will be a saddle-point and we can reduce one saddle-point equation with respect to $\eta_0$. When $\eta_0$ is set to be 0, we have the saddle-point equations of $\eta$ and $x$ respectively as follow

$$0 = -4\pi i x + \frac{i}{2}\log\left[\left(\frac{\cosh(2\pi x) + i\sinh(2\pi\eta)}{\cosh(2\pi x) - i\sinh(2\pi\eta)}\right)^2\right], \tag{4.31a}$$

$$0 = -12\pi i x + 2\pi - 4\pi i\eta + \frac{i}{2}\log\left[\left(\frac{\cosh(2\pi\eta) + i\sinh(2\pi x)}{\cosh(2\pi\eta) - i\sinh(2\pi x)}\right)^2\right]. \tag{4.31b}$$

The first equation can be solved by taking $2\pi i\eta = \arcsin(\sinh(2\pi x))$. By plugging this relation between $\eta$ and $x$ into the second equation, we solve for $x$ and obtain the following saddle-point of $S^{(0)}$

$$x = \frac{1}{4\pi}\log\left(\frac{-1 - 3\sqrt{3}i - \sqrt{-42 + 6\sqrt{3}i}}{4}\right). \tag{4.32}$$

We also explicitly check that $\frac{\partial S^{(0)}}{\partial \eta_0}$ is vanishing when $\eta_0 = 0$ and $x$, $\eta$ take the given saddle-point values. Therefore, $\eta_0 = 0$ is indeed the saddle-point along $\eta_0$ direction as we justified before. By evaluating the $S^{(0)}$ at the saddle point, we recover the hyperbolic volume of the figure eight knot as expected

$$S^{(0)} = -\frac{\mathrm{vol}(4_1)}{2\pi}. \tag{4.33}$$

In order to compare the semiclassical result with the refined volume conjecture (2.1), we should also study the higher-loop corrections. Using the expansion of double-sine functions in (4.18), we can compute the partition function to all orders perturbatively in $b^2$. Here we focus on the order one factor in the expansion

$$Z^{(1)} = \frac{1}{2}\sqrt{-\frac{(2\pi)^3}{\det(\mathrm{Hess}\,S^{(0)})}}\,\mathrm{e}^{S^{(1)}}, \tag{4.34}$$

since this factor is closely related to the one-loop determinant in the 3d gravity calculation. The prefactor comes from the Gaussian integral around the saddle point. The additional factor of $\frac{1}{2}$ appears because the integral is restricted to $P'_0 > 0$, while the minus sign inside the square root originates from the fact that the Gaussian integral has the form $\mathrm{e}^{\frac{1}{b^2}S_0}$. The three factors of $b$ get cancelled against the three $b$'s from the Jacobian in (4.29). We collect all order-one terms in the expansion

$$\mathrm{e}^{S^{(1)}} = \frac{4i\sinh(2\pi x)}{\sinh(\pi(\frac{i}{4} - \frac{\eta_0}{2} \pm x \pm \eta))^{\frac{1}{4}}}, \tag{4.35}$$

which upon inserting the saddlepoint value simplifies to

$$\mathrm{e}^{S^{(1)}} = 4\sqrt{2}i\sinh(2\pi x). \tag{4.36}$$

We then take the Gaussian integral contribution to (4.34) into account, we obtain the order-one correction to the partition function

$$Z^{(1)} = \frac{2\sqrt{2}i\sinh(2\pi x)}{\sqrt{7i + 5\cosh(2\pi x)}\sqrt{6 - 2\cosh(4\pi x) - 5i\cosh(4\pi x)}} = \frac{\sqrt{2}}{3^{\frac{1}{4}}}. \tag{4.37}$$

This result matches with the order-one term in the expression (4.24) with the Reidemeister torsion $\sqrt{\Delta} = \sqrt{3}$ at $P_0 = 0$.

Note that in the refined volume conjecture (2.1), we write the semiclassical expansion of the partition function in terms of the central charge $c$, while we have the $b^2$ expansion in this part of calculation. The central charge $c$ is defined as $c = 1 + 6(b + \frac{1}{b})^2 = 13 + \frac{6}{b^2} + 6b^2$. Therefore, strictly speaking, the one-loop determinant from the gravity calculation is not equal to $Z^{(1)}$. Instead, we need to renormalize $Z^{(1)}$ to obtain the one-loop determinant

$$Z_{\text{one-loop}} = Z^{(1)} e^{\frac{13}{12\pi} \text{vol}(4_1)}, \tag{4.38}$$

which should be compared with the calculations performed in [3].[23]

## 4.5 Dehn surgery

As final application to the figure eight knot computation, we discuss an example of Dehn surgery. Consider the figure eight knot and excise a small tubular neighborhood around the knot. We can then glue back a torus, but twisted by an $\text{SL}(2,\mathbb{Z})$ element. Such an element is specified by a two coprime integers $(p, q)$ specifying the slope of the meridian (the contractible curve).

The Virasoro TQFT partition function on a solid torus gives simply the vacuum character $\chi_{\mathbb{1}}$ in the appropriate channel, while it gives a generic Virasoro character $\chi_P$ with the inclusion of a Wilson line of momentum $P$. We can write[24]

$$Z_{\text{Vir}}(4_1, P_0) = \langle Z_{\text{Vir}}(4_1^\circ) | \chi_{P_0} \rangle, \tag{4.39}$$

where $4_1^\circ$ is the figure eight knot complement with a tubular neighborhood around the knot removed and we emphasize the $P_0$-dependence of the Virasoro TQFT partition function.

Thus the partition function of a manifold obtained by Dehn surgery from the figure eight knot is given by

$$Z_{\text{Vir}}(4_1(p, q)) = \langle Z_{\text{Vir}}(4_1^\circ) | \mathbb{U}(p, q) | \chi_{\text{vac}} \rangle \tag{4.40}$$

$$= \int_0^\infty dP \, \mathbb{U}(p, q)_{\mathbb{1}, P} \langle Z_{\text{Vir}}(4_1^\circ) | \chi_P \rangle \tag{4.41}$$

$$= \int_0^\infty dP \, \mathbb{U}(p, q)_{\mathbb{1}, P} \, Z_{\text{Vir}}(4_1, P), \tag{4.42}$$

where $\mathbb{U}(p, q)$ is the representation of the $\text{SL}(2, \mathbb{Z})$ modular transformation on the Virasoro characters. It takes the explicit form (see e.g. [53])

$$\mathbb{U}(p, q)_{\mathbb{1}, P} = \varepsilon(p, q) \sqrt{\frac{8}{q}} \, e^{-\frac{2\pi i}{q}(p^* \frac{Q^2}{4} - pP^2)} \left( \cosh\left(\frac{2QP\pi}{q}\right) - e^{\frac{2\pi i p^*}{q}} \cosh\left(\frac{2\hat{Q}P\pi}{q}\right) \right). \tag{4.43}$$

Here $\hat{Q} = b - b^{-1}$, $\varepsilon(p, q)$ is a $P$-independent 24-th root of unity coming from the transformation behaviour of the Dedekind $\eta$-function and $p^*$ is the modular inverse of $p$, $pp^* \equiv 1 \mod q$. This leaves an ambiguity in the expression which can be absorbed in the framing ambiguity. For the figure eight knot, we should also notice that because of amphichirality, the Dehn surgeries $(p, q)$ and $(-p, q)$ are equivalent and we can focus on $p, q \geq 0$.

It is in particular simple to evaluate the hyperbolic volume of this class of manifolds via saddle point approximation. Set $P = \frac{\eta}{b}$ as before. Then the action is

$$S = \text{vol}(4_1, \eta) + \frac{\pi^2 i}{q}(p^* - 4p\eta^2) \pm \frac{4\pi^2 \eta}{q}. \tag{4.44}$$

---

[23]The computation in [3] is not directly applicable to the figure eight knot case because of the presence of the cusp, in which case the relevant Kleinian groups has parabolic elements.

[24]As explained in [1], the normalization of the inner product on the torus is somewhat ambiguous, but this ambiguity will cancel out of the calculation.

Table 2: The volumes of manifolds obtained from Dehn surgery from the figure eight knot. Zero entries indicate that the corresponding manifolds do not admit a hyperbolic metric. The other two exceptional cases that do not admit a hyperbolic metric are $(p, q) = (1, 0)$ and $(0, 1)$, see also [54, Theorem 4.7].

| $p$ \ $q$ | 1 | 2 | 3 | 4 | 5 | 6 | 7 | 8 | 9 |
|---|---|---|---|---|---|---|---|---|---|
| 1 | 0 | 1.3985 | 1.7320 | 1.8581 | 1.9186 | 1.9521 | 1.9725 | 1.9858 | 1.9950 |
| 2 | 0 | | 1.7371 | | 1.9195 | | 1.9727 | | 1.9951 |
| 3 | 0 | 1.4407 | | 1.8634 | 1.9210 | | 1.9732 | 1.9862 | |
| 4 | 0 | | 1.7571 | | 1.9231 | | 1.9738 | | 1.9955 |
| 5 | 0.9813 | 1.5295 | 1.7714 | 1.8735 | | 1.9557 | 1.9745 | 1.9870 | 1.9958 |
| 6 | 1.2845 | | | | 1.9287 | | 1.9754 | | |
| 7 | 1.4638 | 1.6496 | 1.8058 | 1.8871 | 1.9321 | 1.9591 | | 1.9882 | 1.9965 |
| 8 | 1.5832 | | 1.8243 | | 1.9358 | | 1.9776 | | 1.9970 |
| 9 | 1.6678 | 1.7521 | | 1.9027 | 1.9397 | | 1.9789 | 1.9897 | |

Since we focus on the volume, we can omit the purely imaginary part involving $p^*$. The sign choice of the last term is also immaterial, since we can send $\eta \to -\eta$. We hence find that

$$\text{vol}(4_1(p, q)) = \text{Re}\left(\text{vol}(4_1, \eta) + \frac{4\pi^2\eta(1 - pi\eta)}{q}\right)\Bigg|_{\eta = \eta^*}, \tag{4.45}$$

where we plug in the saddle-point value $\eta^*$ and the volume is given by (4.26). The saddle-point equation is transcendental and doesn't admit a closed form solution. However, it is straightforward to compute the volumes of various examples numerically, see Table 2. We compared them to the volumes as computed by the program SnapPy. It is also simple to compute the volumes in a large $p$ and $q$ expansion, since for large $p$ or $q$, the saddle point $\eta^* \to 0$ and the volume converges to the volume of the figure eight knot. We find to the first few orders

$$\text{vol}(4_1(p, q)) = \text{vol}(4_1) - \frac{2\sqrt{3}\pi^2}{p^2 + 12q^2} + \frac{4\pi^4(p^4 - 72p^2q^2 + 144q^4)}{\sqrt{3}(p^2 + 12q^2)^4}$$
$$- \frac{8\pi^6(23p^8 - 8904p^6q^2 + 302400p^4q^4 - 1620864p^2q^6 + 767232q^8)}{45\sqrt{3}(p^2 + 12q^2)^7} + \cdots \tag{4.46}$$

The correction to the figure eight knot volume is always negative as required by general theorems about Dehn surgery [54, Theorem 6.5.6.]. This expansion is a known result, see [55]. This case of Dehn surgery exemplifies the existence of accumulation points in the spectrum of three-manifolds. We discussed their implications for the gravitational path integral in our previous paper [1].

# Acknowledgments

We would like to thank Alex Belin, Jeevan Chandra, Tom Hartman, Daniel Jafferis, Diego Liška, Alex Maloney, Baur Mukhametzhanov, Boris Post, Sahand Seifnashri, Steve Shenker, Julian Sonner, Jörg Teschner and Ka Ho Wong for useful discussions.

**Funding information** While at the IAS, L.E. was supported by the grant DE-SC0009988 from the U.S. Department of Energy. This material is based upon work supported by the U.S. Department of Energy, Office of Science, Office of High Energy Physics of U.S. Department of Energy under grant Contract Number DE-SC0012567 (High Energy Theory research), DOE Early Career Award DE-SC0021886 and the Packard Foundation Award in Quantum Black Holes and Quantum Computation.

# A Semiclassical expansion of the double sine function

In this appendix, we will derive the semiclassical expansion of the double sine function (4.18). We start from the integral representation

$$\log S_b(x_0 + x) = \frac{1}{4} \int_{\mathbb{R}+i0^+} \frac{\mathrm{d}t}{t} \frac{\sinh\left((\frac{Q}{2} - x - x_0)t\right)}{\sinh(\frac{bt}{2})\sinh(\frac{t}{2b})} \tag{A.1}$$

$$= \frac{1}{4} \int_{\mathbb{R}+i0^+} \frac{\mathrm{d}t}{t} \frac{e^{(\frac{b^2+1}{2} - bx - bx_0)t} - e^{-(\frac{b^2+1}{2} - bx - bx_0)t}}{(e^{\frac{b^2 t}{2}} - e^{-\frac{b^2 t}{2}})\sinh(\frac{t}{2})} \tag{A.2}$$

$$= \frac{1}{4} \int_{(\mathbb{R}+i0^+)\cup(\mathbb{R}+i0^-)} \frac{\mathrm{d}t}{t} \frac{e^{(b^2+\frac{1}{2} - bx - bx_0)t}}{(e^{b^2 t} - 1)\sinh(\frac{t}{2})}. \tag{A.3}$$

Here we rescaled $t$ and put $t \to -t$ in the second expression to have the integrand have the same form. We can now use the definition of the Bernoulli polynomials and get as formal expansion

$$\log S_b(x_0 + x) = \sum_{n=0}^{\infty} \frac{b^{2n-2}}{4n!} B_n(1 - \tfrac{x}{b}) \int_{(\mathbb{R}+i0^+)\cup(\mathbb{R}+i0^-)} \mathrm{d}t \frac{t^{n-2} e^{(\frac{1}{2} - bx_0)t}}{\sinh(\frac{t}{2})}. \tag{A.4}$$

The remaining integral can be computed for example by pulling off the contour off and summing over the residues at $t = 2\pi i m$. This gives

$$\int_{\mathbb{R}+i0^+} \mathrm{d}t \frac{t^{n-2} e^{(\frac{1}{2} - bx_0)t}}{\sinh(\frac{t}{2})} = \sum_{m=1}^{\infty} 2(2\pi i m)^{n-2} e^{-2\pi i m b x_0} = 2(2\pi i)^{n-2} \mathrm{Li}_{2-n}(e^{-2\pi i b x_0}). \tag{A.5}$$

We similarly evaluate the contribution from the other contour $\mathbb{R} + i0^-$ which then recovers (4.18).

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
