# Peer review of "d gravity from Virasoro TQFT: Holography, wormholes and knots"

_SciPost Physics, doi:SciPost Phys. 17, 134 (2024)_

## Round 1 · Author Response

We would like to thank the referee for their careful reading of the manuscript and for their feedback on the paper. Here we respond to their questions and comments: - Thank you, we have added footnote 3 clarifying our notation for $\Sigma_{g,n}$. - Yes, the existence of mutations means that the Virasoro TQFT partition function is not always capable of distinguishing topologically distinct hyperbolic three-manifolds. The question you raise has to do with the sum over topologies in the gravitational path integral, which we mostly do not address in this paper. In principle it is expected that, given some fixed boundary conditions, one should (at least) sum over all hyperbolic three-manifolds in order for the boundary dual to solve the CFT crossing equations, but this question is beyond the scope of this work. Although the VTQFT partition function of some hyperbolic three-manifolds may coincide, let us note that this does not lead to the vacuum being counted more than once, since such manifolds give contributions supported entirely above the black hole threshold. - In situations where the boundary surfaces have moduli, the answer to the question of which hyperbolic three-manifold consistent with those boundary conditions gives the dominant contribution in the semiclassical limit will in general depend on the moduli; there will be phase transitions as the moduli are varied. In the simpler situation like the ones the referee mentions where the boundary surfaces do not have moduli, the answer to this question depends on a careful analysis of the Virasoro TQFT partition functions in the semiclassical limit. We are not aware of a general heuristic that determines the dominant topology. - The question of the holographic interpretation of the figure-eight knot complement (and indeed of hyperbolic knot complements in general) is an interesting one, since knot complements do not a priori have asymptotic boundaries on which the dual CFT can live. However, such links play an important role as building blocks of more general three-manifolds (including those with asymptotic boundaries), since a general three-manifold may be obtained by performing Dehn surgery on such links embedded in $S^3$. They will hence play a central role in implementing the sum over topologies in the gravity path integral. We worked through the example of Dehn surgery on the figure-eight knot in section 4.5, which furnishes a family of hyperbolic three-manifolds whose volume accumulates to that of the figure-eight knot complement (although these examples do not have asymptotic boundaries).

---

## Round 1 · List of Changes

- We have added footnote 3 clarifying our notation for $\Sigma_{g,n}$.

---

## Editorial Decision

published